# Structured Matrix Basis for Multivariate Time Series Forecasting with Interpretable Dynamics

**Xiaodan Chen**[1]**, Xiucheng Li**[2] **(✉), Xinyang Chen**[2]**, Zhijun Li**[1] **(✉)**
[1] School of Computer Science and Technology, Harbin Institute of Technology
[2] School of Computer Science and Technology, Harbin Institute of Technology, Shenzhen
`{21B303004@stu., lixiucheng@, chenxinyang@, lizhijun_os@}hit.edu.cn`

## Abstract

Multivariate time series forecasting is of central importance in modern intelligent decision systems. The dynamics of multivariate time series are jointly characterized by temporal dependencies and spatial correlations. Hence, it is equally important to build the forecasting models from both perspectives. The real-world multivariate time series data often presents spatial correlations that show structures and evolve dynamically. To capture such dynamic spatial structures, the existing forecasting approaches often rely on a two-stage learning process (learning dynamic series representations and then generating spatial structures), which is sensitive to the small time-window input data and has high variance. To address this, we propose a novel forecasting model with a structured matrix basis. At its core is a dynamic spatial structure generation function whose output space is well-constrained and the generated structures have lower variance, meanwhile, it is more expressive and can offer interpretable dynamics. This is achieved via a novel structured parameterization and imposing structure regularization on the matrix basis. The resulting forecasting model can achieve up to $8.5\%$ improvements over the existing methods on six benchmark datasets, and meanwhile, it enables us to gain insights into the dynamics of underlying systems.

## 1 Introduction

Multivariate time series forecasting plays a pivotal role in a wide range of fields, such as traffic flow management, electricity consumption, and weather prediction. Multivariate time series data records quantities of interest from $N$ series spanning over $T$ time steps, and its underlying dynamics are jointly characterized by the temporal correlations (intra-series dependencies) and spatial structures (inter-series dependencies). Inspired by the advancements in Natural Language Processing, substantial research has been proposed to apply RNNs and Transformers to capture the underlying temporal dependencies [41, 32, 44, 45, 30]. Besides, the convolution paradigm also exhibits promising temporal correlation modeling capability and excels in long-term forecasting [33, 36, 29].

As the underlying dynamics are jointly described by the intra- and inter-correlations, it is equally important to explore the spatial structures for an ideal forecasting model design. Many proposals employ dense connection to capture the spatial correlations implicitly [31, 24]. However, the dense connection lacks clear structures and is prone to introduce noise from uncorrelated spatial dimensions. The development of Graph Neural Networks (GNNs) [18, 17, 7] offers an effective solution to model non-Euclidean structure data. DCRNN [22] builds graphs based on spatial proximity and conducts graph convolution to capture spatial correlations for traffic forecasting. To apply GNNs to more general scenarios where graph structures are unavailable, the forecasting methods propose to learn the graph adaptively through learnable node embeddings [37, 1, 38, 15], which significantly enhances the forecasting performance.

38th Conference on Neural Information Processing Systems (NeurIPS 2024).

Despite the progress achieved, the spatial structures remain static across time steps for the aforementioned forecasting methods, which may not reflect the actual inter-series correlation. Because in many scenarios, the spatial correlations are also changing dynamically, for example, the traffic speeds of certain road segments manifest correlation only in peak hours. To relax this restriction, many dynamic graph-based methods have been proposed [43, 39, 42, 28, 34] to learn the spatial structures within a short time window. The intuition is that the spatial correlations in real-world applications often evolve continuously and tend to be stable over a short period of time. The general idea is to learn each series a dynamic representation via a nonlinear transformation $f_{\text{dynm}}$ by taking as input the current time window data, and it then generates the spatial structures by pairwise interacting the dynamic representations via a transformation $f_{\text{pair}}$, the composition of two transformations forms the spatial structure generation function $f_{\text{spatial}} = f_{\text{pair}} \circ f_{\text{dynm}}$. The dynamic representation function $f_{\text{dynm}}$ is often implemented as MLP ([43, 42, 28, 34]) or RNN ([39]) whereas $f_{\text{pair}}$ is mostly instantiated by attention mechanism or inner product. One severe issue of these methods is that the output space of $f_{\text{dynm}}$ is not well constrained and unbounded, this makes the learned dynamic representations very sensitive to the short time window input data and the unboundedness will be exaggerated by the inner product operation in $f_{\text{pair}}$, which will lead to the outputs of $f_{\text{spatial}}$ fluctuate drastically and have high variance. The issue becomes even more severe in the presence of anomaly patterns. To reduce the variance, TPGNN [26] first learns a static spatial structure $\mathbf{A}$ (adjacency matrix) and then generates the dynamic spatial structures with a matrix polynomial $\sum_{m=1}^{M} \alpha_m \mathbf{A}^m$ where $\alpha_m$ is determined by the timestamps of the current time window. However, it has two drawbacks: 1) the coefficient $\alpha_m$ solely depends on the timestamps and cannot adapt to the current window data, and 2) the matrix power basis $\mathbf{A}, \mathbf{A}^2, \ldots, \mathbf{A}^M$ is overly restricted and has limited expressive capability. In addition, the existing forecasting methods often lack interpretable dynamics.

In this paper, we propose a dynamic multivariate time series forecasting model with a structured matrix basis. Instead of relying on the two-stage spatial structure learning process (learning dynamic series representations and then generating spatial structures), we directly parameterize $f_{\text{spatial}}$ with a learnable matrix basis $\mathbf{B}_1, \mathbf{B}_2, \ldots, \mathbf{B}_M$ and represent any spatial structure with a convex combination $\sum_{m=1}^{M} \alpha_m \mathbf{B}_m$, $\alpha_m \geq 0$ and $\sum_{m=1}^{M} \alpha_m = 1$. To learn the matrix basis effectively, we propose a novel structured matrix parameterization method and impose structure regularization on the basis to enhance parameter efficiency and reduce complexity. In contrast to the two-stage spatial structure learning methods, the output space of our proposed $f_{\text{spatial}}$ is well constrained. Consequently, the generated spatial structures have lower variance and the resulting model is easier to learn. In comparison to TPGNN, our matrix basis is more expressive since it is not limited by the matrix power constraint; the coefficient $\boldsymbol{\alpha}$ can also be computed adaptively via the interaction of current time-window data and the basis. In addition, the coefficient $\boldsymbol{\alpha}$ offers a fashion to track the spatial structure evolution and enables us to gain insights into the underlying dynamics. Thus, the resulting model is more interpretable.

In summary, our proposed $f_{\text{spatial}}$ has the following appealing properties: 1) lower variance and easier to learn, 2) it is more expressive, and 3) it can yield more interpretable results. This is achieved through a novel structured matrix parameterization and structure regularization. By integrating $f_{\text{spatial}}$ into the forecasting framework, we evaluate the efficacy of the proposed method on six benchmark datasets, it achieves up to $8.5\%$ improvements over existing forecasting methods across various prediction lengths and can also offer interpretable dynamics.

## 2  Related Work

**Temporal Dependency Modeling**  Early deep sequential methods adopt recurrent neural networks to capture nonlinear temporal dynamics [41, 32, 5]. Motivated by the wide receptive fields of attention mechanism, various Transformer-based methods have been developed to capture the long-term temporal dependencies in forecasting. To reduce the quadratic complexity of the vanilla attention, LogTrans [21], Informer [44], Autoformer [35], and FEDformer [45] have been proposed successively. Non-stationary Transformer [27] attempts to mitigate the difficulties caused by non-stationarity in modeling temporal correlation. PatchTST [30] explores the strategies of patch-level semantic modeling and channel-independence. VQ-MTM [13] explores the well-defined semantic units for the Transformer architecture in time series modeling. In state-space models, the transition matrix is employed to model long-term temporal dependencies, and their recent representative works include Hippo [10], LSSL [11]. To reduce the computational complexity, S4 [12] and Mamba [9]

have been proposed successively. The convolution-based methods have also shown promising results in time series modeling. MICN [33] explores isometric convolution to capture non-local temporal patterns. TimesNet [36] reshapes 1D signals into 2D by aligning according to inherent multiple periodicity and it then employs the 2D kernels to capture both intraperiod- and interperiod-variations. ModernTCN [29] utilizes large kernels to model long-term dependencies. In addition, the Fourier basis parameterization has also been proposed to model long-term dependencies in time series imputation [25].

**Static Spatial Correlation Modeling** The spatial correlation plays an equally important role in time series modeling. DeepAR [31] and Pyraformer [24] employ dense connection to model the spatial dependencies. However, the dense connection fails to explore the underlying structure and may introduce noises from unrelated series. The advancement of GNNs offers an effective way to model non-Euclidean structure data. DCRNN [22] constructs the graph by using spatial proximity and proposes to fuse the spatial information via graph convolution operation, and thus it is only applicable when the underlying graph structures are easily accessible. To sidestep this limitation, the adaptive GNNs-based methods [1, 38, 15, 14, 16] propose to learn each series a representation and then generate the spatial structure via the interaction of the representations. BiTGraph [4] further develops the method to account for missing patterns in message passing to handle the time series with missing values. However, these approaches implicitly assume the series representations are shared across the entire time steps, and hence the underlying spatial graphs remain static over time.

**Dynamic Spatial Correlation Modeling** In real-world applications, the inter-series correlations or spatial structures are often evolving dynamically. To adapt to these scenarios, many dynamic spatial structure methods are proposed, which share a similar two-stage spatial structure learning process as the static adaptive GNNs-based methods (as discussed in Section 1). The difference is that the series representations are generated by conditioning a small time window rather than the entire timeline data, and hence, the underlying spatial structures can change dynamically over time. GMAN [43], iTransformer [28], Crossformer [42], and Card [34] implement the dynamic node representation function via MLP, whereas ESG [39] adopts the RNN. As the output spaces of their $f_{\text{spatial}}$ are not well constrained, the learned spatial structures are very sensitive to the change of time-window data and hence have high variance. To reduce the variance, TPGNN [26] proposes to represent the dynamic graphs with matrix polynomial. As the polynomial coefficients are solely determined by the timestamps, such a method fails to utilize the current time-window data. Moreover, the matrix power basis is overly restricted and has weak expressive capability. The proposals [40, 8] attempt to build dynamic graphs by merging temporal and spatial dimensions. Nevertheless, the entanglement of temporal correlations and spatial dependencies makes the models hard to optimize. In addition, the existing multivariate time series forecasting methods also lack interpretability regarding the underlying dynamics.

## 3 Methodology

**Notation** From a generative perspective, the multivariate time series $\mathbf{X} \in \mathbb{R}^{N \times T \times D}$ records a $D$-dimensional physical quantities of interest generated by $N$ series (i.e., sensors or instances) over $T$ time steps. We use $\mathbf{X}^{(n)} \in \mathbb{R}^{T \times D}$ to represent the observations from $n$-th sensor and $\mathbf{X}_t \in \mathbb{R}^{N \times D}$ to indicate the observations at the $t$-th timestamp. The slice notation $\mathbf{X}_{t-H:t} \in \mathbb{R}^{N \times H \times D}$ denotes the values within a window spanning from the time interval $[t-H, t)$. The operator $\text{diag} : \mathbb{R}^{N \times N} \mapsto \mathbb{R}^N$ takes the diagonal elements of a square matrix and returns it as a vector, the operator $\text{vec}$ reshapes a matrix or tensor into a vector.

**Overview and Pipeline** Figure 1-(a) presents the architecture of our proposed Sumba (dynamic multivariate time series forecasting with structured matrix basis), which comprises $L$ blocks. Each block contains two primary modules: the Multi-Scale TCN and Dynamic GCN modules. The Multi-Scale TCN module in the $\ell$-th block takes as the input $\mathbf{Z}_{t_0-H:t_0}^{(\ell-1)} \in \mathbb{R}^{N \times H \times D_i^{(\ell-1)}}$ and generates the intermediate representation $\mathbf{Z}'^{(\ell-1)}_{t_0-H:t_0} \in \mathbb{R}^{N \times H \times D_o^{(\ell-1)}}$, which is fed to the Dynamic GCN module to produce $\mathbf{Z}_{t_0-H:t_0}^{(\ell)} \in \mathbb{R}^{N \times H \times D^{(\ell)}}$. The Multi-Scale TCN captures the temporal dependencies by performing multi-scale temporal convolution operation in a channel-independent manner, we choose the kernel sizes $1 \times 2$, $1 \times 3$, $1 \times 6$, and $1 \times 7$ in this paper. The Dynamic GCN module comprises two functions, namely, the spatial structure generation function $f_{\text{spatial}}$ and graph convolution function $f_{\text{gcn}}$. Our proposed $f_{\text{spatial}}$ generates the dynamic spatial structure $\mathbf{A}_t \in \mathbb{R}^{N \times N}$ (adjacency matrix)

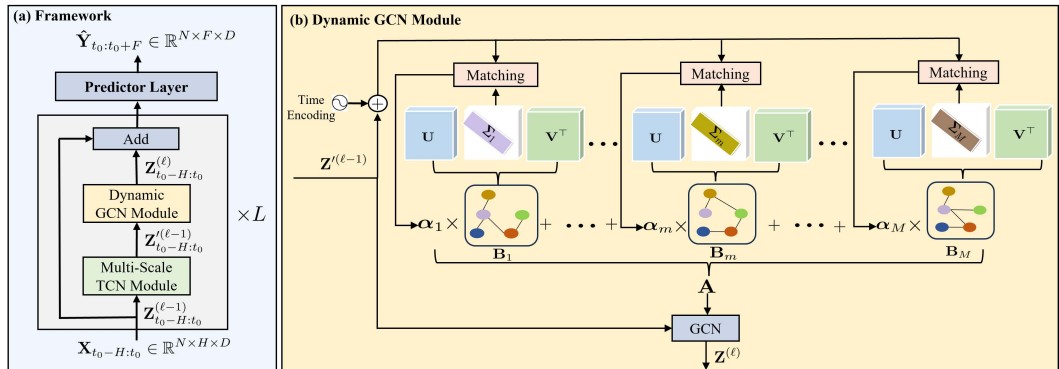

Figure 1: **(a)** The framework of our proposed Sumba. **(b)** the detailed structure of the Dynamic GCN module.

by conditioning on the intermediate representation $\mathbf{Z}_t'^{(\ell-1)} \in \mathbb{R}^{N \times D_o^{(\ell-1)}}$ at time step $t \in [t_0 - H, t_0)$, which is obtained by transforming the current time-window data $\mathbf{X}_{t_0-H:t_0}$. Given the generated dynamic graph $\mathbf{A}_t$, $f_{\mathrm{gcn}}$ further fuses the spatial information of $\mathbf{Z}_t'^{(\ell-1)}$ across different series to yield $\mathbf{Z}_t^{(\ell)}$ by performing the graph convolution operation. Note that the spatial and temporal dimensions are kept unchanged during the entire transformation in the $L$ blocks.

## 3.1 Adaptive Dynamic Spatial Structure Generation

The core layer of our proposed forecasting method is the spatial structure generation function $f_{\mathrm{spatial}}$. It serves to infer the optimal graph structure $\mathbf{A}_t$ that best characterizes the present spatial correlations from the intermediate representation $\mathbf{Z}_t'$ (we drop the layer index $\ell$ to keep the notation uncluttered in this subsection). As mentioned in Section 1, the existing methods all adopt the two-stage learning process, which results in unconstrained output function space and high graph structure variance. To address this, we propose to directly parameterize a learnable matrix basis $\mathcal{B} = \{\mathbf{B}_m\}_{m=1}^M$ of dimension $M$ where $\mathbf{B}_m \in \mathbb{R}^{N \times N}$ and define the spatial structure generation function as

$$f_{\mathrm{spatial}}(\mathbf{Z}_t') \triangleq \sum_{m=1}^M \alpha_{t,m} \mathbf{B}_m, \quad \alpha_{t,m} \geq 0, \sum_{m=1}^M \alpha_{t,m} = 1. \tag{1}$$

Here, we choose the convex combination instead of the linear combination to better control the output space. This is reasonable since the basis $\mathcal{B}$ is free to optimize in the training stage. The intuition behind Eq. 1 is that the basis $\mathcal{B}$ can be shared and optimized globally across different time windows, and to infer the spatial structure dynamically, we only need to adaptively compute the coefficient $\boldsymbol{\alpha}_t \in \mathbb{R}^M$ by conditioning on $\mathbf{Z}_t'$.

However, two challenges remain in adopting the spatial structure generation function in Eq. 1. 1) The number of parameters to be learned equals $MN^2$, which grows quadratically to $N$. As these $M$ matrices lack connection and constraints, learning by brute force will sooner become infeasible even for medium-size $N$. This actually is the reason why the existing methods resort to the two-stage learning process, i.e., learning each series an embedding whose learnable parameters are $ND$ with $D$ being the embedding dimension. 2) Intuitively, the best $\alpha_m$ should be simultaneously determined by $\mathbf{Z}'$ and $\mathbf{B}_m$, but identifying $\mathbf{B}_m$ with $N^2$ parameters will make $\alpha_m$ hard to compute when $N$ is large.

**Structured Parameterization and Regularization** To circumvent the two challenges, we propose to parameterize the basis matrices in a structured manner and impose additional structure regularization on the basis. The idea is to represent each basis matrix $\mathbf{B}_m$ in its SVD (singular value decomposition) factor product form $\mathbf{B}_m = \mathbf{U}_m \Sigma_m \mathbf{V}_m^\top$ and then parameterize the factors $\mathbf{U}_m, \Sigma_m, \mathbf{V}_m$, where $\mathbf{U}_m, \mathbf{V}_m \in \mathbb{R}^{N \times N}$ are orthogonal matrices, and $\Sigma_m$ is a diagonal matrix consisting of the singular values of $\mathbf{B}_m$. **One benefit** of such structured parameterization is that it permits us to establish connections between the basis matrices and impose constraints, and consequently, enhance the parameter efficiency and ease of the model learning. To be specific, we impose the constraint that all $\mathbf{B}_m$ for $m = 1, 2, \ldots, M$ share the same parameterized orthogonal matrices $\mathbf{U}, \mathbf{V}$ and each matrix has its unique $\Sigma_m$, i.e.,

$$\mathcal{B} \triangleq \left\{ \mathbf{U}\Sigma_1 \mathbf{V}^\top, \mathbf{U}\Sigma_2 \mathbf{V}^\top, \ldots, \mathbf{U}\Sigma_M \mathbf{V}^\top \right\}. \tag{2}$$

The rationality behind such a choice stems from the geometry interpretation of orthogonal matrix-vector multiplication, i.e., left multiplying a vector by an orthogonal matrix is equivalent to coordinate transformation, which is provided in Appendix A.1. Hence, we implicitly require that all basis matrices share the same pair of coordinate transformations, which can be considered a sort of implicit regularization since it reduces extra freedom and guides the model to find the coordinate transformations ($\mathbf{U}$ and $\mathbf{V}$) that best suit the basis.

**Dynamic Coefficient Generation** Such a parameter sharing regularization mechanism along with the structured parameterization also brings **another benefit**, it allows us to treat $\text{diag}(\Sigma_m) \in \mathbb{R}^N$ as a fingerprint to identify each $\mathbf{B}_m$. Hence, we can compute the $\boldsymbol{\alpha}_t$ to infer the dynamic spatial structure by simultaneously conditioning on $\mathbf{Z}'_t$ and $\text{diag}(\Sigma_m)$. To this end, we design an adaptive matching module that takes as input $\mathbf{Z}'_t \in \mathbb{R}^{N \times D_o}$ and $[\text{diag}(\Sigma_1), \text{diag}(\Sigma_2), \ldots, \text{diag}(\Sigma_M)]$, and it yields the coefficient $\boldsymbol{\alpha}_t$ at the $t$ time step, as follows.

$$
\begin{aligned}
\mathbf{D} &\triangleq [\text{diag}(\Sigma_1), \text{diag}(\Sigma_2), \ldots, \text{diag}(\Sigma_M)] \in \mathbb{R}^{N \times M} \\
\mathbf{z} &\triangleq \text{vec}(\mathbf{Z}'_t) + \text{TimeEncoding}(t) \qquad\qquad \in \mathbb{R}^{ND_o} \\
\boldsymbol{\alpha}_t &= \text{softmax}\left( (\mathbf{W}_d \mathbf{D})^\top \mathbf{W}_z \mathbf{z} / \sqrt{d} \right) \qquad \in \mathbb{R}^M
\end{aligned}
\tag{3}
$$

where $\text{TimeEncoding}$ is the timestamp encoding function, $\mathbf{W}_d$ and $\mathbf{W}_z$ are used to match the dimension, i.e., mapping $\mathbf{D}$ and $\mathbf{z}$ into the $\mathbb{R}^d$ space. Given the structured basis in Eq. 2, each $\boldsymbol{\alpha}_t$ represents a dynamic spatial structure or graph at step $t$ as

$$
\mathbf{A}_t = f_{\text{spatial}}(\mathbf{Z}'_t) = \sum_{m=1}^{M} \alpha_{t,m} \mathbf{U} \Sigma_m \mathbf{V}^\top.
\tag{4}
$$

Given $\mathbf{A}_t$ at each time step, we can perform graph convolution operation to aggregate information from the spatial dimension. The process is illustrated in Figure 1-(b). The output space of $f_{\text{spatial}}$ is well constrained on the premise that $\Sigma_m$ are well bounded, which is given by the following theorem.

**Theorem 3.1.** *The output space of $f_{\text{spatial}}$ in Eq. 4 is bounded by the sum of the maximum of $\Sigma_m$ (the maximum singular value of $\mathbf{B}_m$) for $m = 1, 2, \ldots, M$ in terms of the $\ell_2$ norm i.e., $\|f_{\text{spatial}}\|_2 \leq \sum_{m=1}^{M} \max(\Sigma_m)$.*

The proof is presented in Appendix A.2 and Theorem 3.1 states that the variance of the learned structures is controllable by restricting the maximum value of $\Sigma_m$, which is easy to achieve since $\Sigma_m$ can simply be parameterized by a vector with nonnegative values. Besides, by tracking the change of $\boldsymbol{\alpha}_t$ over time, our proposed method enables us to gain insight into the underlying dynamics of the system, and thus offers additional interpretability, as we will show in Section 4.4.

**Orthogonality** To impose the orthogonal constraint, one may attempt to apply an orthogonality penalty, i.e., by adding the penalty term $\|\mathbf{U}^\top \mathbf{U} - \mathbf{I}\| + \|\mathbf{V}^\top \mathbf{V} - \mathbf{I}\|$ to the optimized objective. However, such a hard penalty cannot guarantee genuine orthogonality and the extra penalty term also increases the learning difficulty. Hence, rather than relying on the hard penalty, we opt for the orthogonal parameterization. In particular, we restrict our attention to the special orthogonal group

$$
\text{SO}(N) \triangleq \left\{ \mathbf{A} \in \mathbb{R}^{N \times N} \mid \mathbf{A}^\top \mathbf{A} = \mathbf{I}, \det(\mathbf{A}) = 1 \right\},
\tag{5}
$$

which are flexible enough to represent the coordinate transformation. A nice property of $\text{SO}(N)$ is that it is both a compact Lie group and a smooth manifold [6]. The tangent space of the Lie group at the identity forms a vector space equipped with a Lie bracket operation, namely, its Lie algebra. The Lie algebra of $\text{SO}(N)$ is the set of all skew-symmetric matrices, denoted by $\mathfrak{so}(N)$, i.e.,

$$
\mathfrak{so}(N) \triangleq \left\{ \mathbf{A} \in \mathbb{R}^{N \times N} \mid \mathbf{A}^\top = -\mathbf{A} \right\}.
\tag{6}
$$

In differentiable manifold and Lie group, a well-known result is that the matrix exponential $\exp : \mathfrak{so} \mapsto \text{SO}$ establishes the connection between a Lie group and its Lie algebra, i.e., for any $\mathfrak{g} \in \mathfrak{so}(N)$ we have $\exp(\mathfrak{g}) \in \text{SO}(N)$. The matrix exponential is defined as

$$
\exp(\mathbf{A}) \triangleq \sum_{k=0}^{\infty} \frac{1}{k} \mathbf{A}^k = \mathbf{I} + \mathbf{A} + \frac{1}{2} \mathbf{A}^2 + \ldots
\tag{7}
$$

The computation of the matrix exponential map is costly but for the special orthogonal group it has a cheap first order approximation, also known as the Clay map [3, 2],

$$\phi(\mathbf{A}) \triangleq \left(\mathbf{I} + \frac{1}{2}\mathbf{A}\right)\left(\mathbf{I} - \frac{1}{2}\mathbf{A}\right)^{-1}. \tag{8}$$

The Clay map can be implemented in parallel by the Gaussian elimination algorithm in a numerically stable way.

**Low Rank Approximation** By using the Clay map, the number of parameters required to parameterize $\mathbf{U}$ and $\mathbf{V}$ is $N(N-1)$ (two skew-symmetric matrices $N(N-1)/2 + N(N-1)/2$). Thus the total number of parameters required by the basis in Eq. 2 is $N(N-1) + MN$. To further reduce the parameter count, we can apply the low rank approximation. In real-world applications, the spatial structures (adjacency matrices) of multivariate time series are often low-ranked. Suppose the rank is $K$, we can only preserve the first $K$ columns of $\mathbf{U}$ and $\mathbf{V}$ as well as parameterize each $\Sigma_m$ with a length $K$ vector, which leads to a parameter count upper bound $NK + MK$. Since $K$ is often much less than $N$ in practice, the low rank approximation can enhance both the parameter and computation efficiency when $N$ is large.

## 3.2 Hierarchical Architecture

By stacking $L$ blocks, the Sumba significantly enhances its capability to model temporal correlations and spatial dependencies effectively. We initialize $\mathbf{Z}^{(0)}$ with the original input $\mathbf{X} \in \mathbb{R}^{N \times H \times D}$. The output of $L$-th block $\mathbf{Z}^{(L)} \in \mathbb{R}^{N \times H \times D^{(L)}}$ produces the multi-step prediction $\hat{\mathbf{Y}}_{t_0:t_0+F}$ through a linear transformation. The model is optimized by minimizing mean absolute error (MAE):

$$\mathcal{L}_{\mathrm{MAE}}(\mathbf{Y}_{t_0:t_0+F}, \hat{\mathbf{Y}}_{t_0:t_0+F}) \triangleq \frac{\sum_{n=1}^{N}\sum_{t=t_0}^{t_0+F-1}|\hat{y}_t^{(n)} - y_t^{(n)}|}{N \times F}. \tag{9}$$

# 4 Experiments

In this section, we evaluate our approach Sumba against 15 time series forecasting methods on six benchmark datasets (Section 4.2 and Appendix C). The ablation studies are presented in Section 4.3. We demonstrate the interpretability of our method with case studies in Section 4.4. The sensitivity of hyperparameters is provided in Section 4.5 and Appendix D; the computational cost is empirically studied in Appendix E. The code of Sumba is available at: https://github.com/chenxiaodanhit/Sumba/.

## 4.1 Experimental Setup

**Datasets** We conduct experiments on six commonly adopted public datasets including: (1) **Electricity** [44] contains hourly electricity consumption of 321 clients. (2) **Weather** [35] includes 21 meteorological factors collected every 10 minutes from the weather station of the Max Planck Biogeochemistry Institute. (3) **PEMS** [23] records traffic data of 358 variates in California sampled every 5 minutes. (4) **ETTh2** [44] contains hourly data from 7 electricity transformers. (5) **Traffic** [35] measures the hourly road occupancy rates of 862 sensors on San Francisco Bay area freeways. (6) **Solar-Energy** [20] records the solar power production, which is sampled every 10 minutes from 137 PV plants.

**Baselines** We compare our method with the following baselines: (1) TCN-based methods: MICN [33], ModernTCN [29]; (2) Transformer-based methods: PatchTST [30], FEDformer [45], Autoformer [35], Reformer [19]; (3) Static graphs-based methods: MTGNN [38], MegaCRN [15]; (4) Dynamic graphs-based methods: iTransformer [28], Crossformer [42], Card [34], ESG [39], TPGNN [26], FourierGNN [40]; (5) Structured State Space model: S4 [12]. More details of baselines are provided in Appendix B.

**Implementation details** The number of blocks $L$ of Sumba is set to 3, the dimension of structured basis $M$ is set to 5, and the rank $K$ is set to $\min(N, 30)$ in all our experiments. The batch size is 32, the learning rate is 0.0001. We split the datasets into training, validation, and test datasets with the ratio 0.6/0.2/0.2 chronologically. The future window size $F$ is set to 3, 6, 12, and 24 for all

Table 1: The forecasting results with prediction horizons of 3 and 6 on Electricity, Weather, etc.

| Method | Electricity | | | Weather | | |
|---|---|---|---|---|---|---|
| (F = 3) | MAE | RMSE | MAPE(%) | MAE | RMSE | MAPE(%) |
| MICN [33] | $0.217 \pm 0.001$ | $0.323 \pm 0.001$ | $2.380 \pm 0.012$ | $0.0812 \pm 0.001$ | $0.217 \pm 0.002$ | $1.563 \pm 0.017$ |
| ModernTCN [29] | $0.172 \pm 0.001$ | $0.259 \pm 0.001$ | $1.629 \pm 0.008$ | $0.0727 \pm 0.000$ | $0.213 \pm 0.001$ | $1.695 \pm 0.056$ |
| PatchTST [30] | $0.173 \pm 0.001$ | $0.266 \pm 0.002$ | $1.359 \pm 0.006$ | $0.0642 \pm 0.000$ | $0.214 \pm 0.000$ | $2.511 \pm 0.223$ |
| FEDformer [45] | $0.268 \pm 0.002$ | $0.380 \pm 0.002$ | $2.454 \pm 0.011$ | $0.236 \pm 0.001$ | $0.364 \pm 0.001$ | $4.832 \pm 0.039$ |
| Autoformer [35] | $0.271 \pm 0.003$ | $0.383 \pm 0.001$ | $2.543 \pm 0.005$ | $0.238 \pm 0.001$ | $0.386 \pm 0.002$ | $7.133 \pm 0.693$ |
| Reformer [19] | $0.324 \pm 0.002$ | $0.452 \pm 0.002$ | $2.921 \pm 0.010$ | $0.0821 \pm 0.001$ | $0.220 \pm 0.002$ | $1.533 \pm 0.022$ |
| S4 [12] | $0.331 \pm 0.002$ | $0.479 \pm 0.002$ | $2.792 \pm 0.015$ | $0.0921 \pm 0.001$ | $0.292 \pm 0.003$ | $4.782 \pm 0.055$ |
| MTGNN [38] | $0.153 \pm 0.001$ | $0.245 \pm 0.002$ | $\underline{1.340 \pm 0.004}$ | $0.0693 \pm 0.000$ | $0.219 \pm 0.001$ | $1.624 \pm 0.018$ |
| MegaCRN [15] | $0.272 \pm 0.001$ | $0.394 \pm 0.002$ | $\underline{2.429 \pm 0.026}$ | $0.0674 \pm 0.000$ | $0.215 \pm 0.001$ | $1.458 \pm 0.032$ |
| iTransformer [28] | $0.158 \pm 0.001$ | $0.248 \pm 0.000$ | $1.472 \pm 0.007$ | $0.0644 \pm 0.000$ | $0.216 \pm 0.001$ | $1.628 \pm 0.026$ |
| CrossFormer [42] | $0.156 \pm 0.000$ | $0.246 \pm 0.001$ | $1.370 \pm 0.008$ | $0.0643 \pm 0.002$ | $\underline{0.210 \pm 0.003}$ | $\underline{1.417 \pm 0.035}$ |
| Card [34] | $\underline{0.152 \pm 0.001}$ | $\underline{0.244 \pm 0.001}$ | $1.460 \pm 0.013$ | $\underline{0.0641 \pm 0.000}$ | $0.215 \pm 0.001$ | $\underline{1.523 \pm 0.007}$ |
| ESG [39] | $0.153 \pm 0.002$ | $0.247 \pm 0.001$ | $1.434 \pm 0.005$ | $0.0753 \pm 0.000$ | $0.215 \pm 0.001$ | $1.837 \pm 0.072$ |
| FourierGNN [40] | $0.213 \pm 0.002$ | $0.327 \pm 0.003$ | $1.619 \pm 0.006$ | $0.0859 \pm 0.000$ | $0.229 \pm 0.002$ | $1.732 \pm 0.031$ |
| TPGNN [26] | $0.206 \pm 0.003$ | $0.324 \pm 0.001$ | $1.677 \pm 0.009$ | $0.0652 \pm 0.000$ | $0.219 \pm 0.001$ | $1.538 \pm 0.024$ |
| Sumba | $\mathbf{0.148 \pm 0.001}$ | $\mathbf{0.237 \pm 0.000}$ | $\mathbf{1.331 \pm 0.005}$ | $\mathbf{0.0587 \pm 0.000}$ | $\mathbf{0.208 \pm 0.000}$ | $\mathbf{1.381 \pm 0.011}$ |

| Method | ETTh2 | | | Traffic | | |
|---|---|---|---|---|---|---|
| (F = 3) | MAE | RMSE | MAPE(%) | MAE | RMSE | MAPE(%) |
| MICN [33] | $0.225 \pm 0.002$ | $0.341 \pm 0.001$ | $1.126 \pm 0.008$ | $0.253 \pm 0.001$ | $0.577 \pm 0.001$ | $2.794 \pm 0.013$ |
| ModernTCN [29] | $0.242 \pm 0.002$ | $0.340 \pm 0.002$ | $1.153 \pm 0.013$ | $0.239 \pm 0.002$ | $0.562 \pm 0.002$ | $3.210 \pm 0.036$ |
| PatchTST [30] | $\underline{0.182 \pm 0.001}$ | $\mathbf{0.285 \pm 0.001}$ | $\underline{0.854 \pm 0.003}$ | $0.226 \pm 0.001$ | $0.532 \pm 0.002$ | $2.440 \pm 0.009$ |
| FEDformer [45] | $0.332 \pm 0.012$ | $0.480 \pm 0.006$ | $1.522 \pm 0.023$ | $0.346 \pm 0.006$ | $0.714 \pm 0.009$ | $4.010 \pm 0.041$ |
| Autoformer [35] | $0.299 \pm 0.003$ | $0.438 \pm 0.004$ | $1.344 \pm 0.017$ | $0.361 \pm 0.003$ | $0.728 \pm 0.005$ | $3.956 \pm 0.026$ |
| Reformer [19] | $0.442 \pm 0.006$ | $0.663 \pm 0.011$ | $2.091 \pm 0.025$ | $0.322 \pm 0.007$ | $0.757 \pm 0.014$ | $3.351 \pm 0.038$ |
| S4 [12] | $0.441 \pm 0.004$ | $0.617 \pm 0.018$ | $2.749 \pm 0.017$ | $0.295 \pm 0.003$ | $0.734 \pm 0.003$ | $2.647 \pm 0.011$ |
| MTGNN [38] | $0.185 \pm 0.001$ | $0.289 \pm 0.001$ | $0.855 \pm 0.004$ | $0.203 \pm 0.002$ | $0.525 \pm 0.003$ | $2.332 \pm 0.008$ |
| MegaCRN [15] | $0.237 \pm 0.002$ | $0.352 \pm 0.001$ | $1.177 \pm 0.005$ | $0.326 \pm 0.002$ | $0.735 \pm 0.007$ | $4.054 \pm 0.028$ |
| iTransformer [28] | $0.188 \pm 0.000$ | $0.294 \pm 0.002$ | $0.906 \pm 0.001$ | $\underline{0.195 \pm 0.001}$ | $\mathbf{0.492 \pm 0.001}$ | $2.155 \pm 0.007$ |
| CrossFormer [42] | $0.206 \pm 0.001$ | $0.325 \pm 0.002$ | $0.927 \pm 0.003$ | $0.198 \pm 0.002$ | $0.602 \pm 0.006$ | $\underline{2.079 \pm 0.021}$ |
| Card [34] | $\underline{0.182 \pm 0.000}$ | $0.287 \pm 0.000$ | $0.869 \pm 0.006$ | $0.197 \pm 0.001$ | $\underline{0.495 \pm 0.001}$ | $2.147 \pm 0.007$ |
| ESG [39] | $0.217 \pm 0.001$ | $0.335 \pm 0.009$ | $0.909 \pm 0.015$ | $0.198 \pm 0.001$ | $0.588 \pm 0.003$ | $2.086 \pm 0.004$ |
| FourierGNN [40] | $0.232 \pm 0.003$ | $0.376 \pm 0.005$ | $1.436 \pm 0.016$ | $0.264 \pm 0.002$ | $0.585 \pm 0.002$ | $2.934 \pm 0.019$ |
| TPGNN [26] | $0.207 \pm 0.003$ | $0.329 \pm 0.002$ | $0.986 \pm 0.013$ | $0.288 \pm 0.003$ | $0.665 \pm 0.004$ | $3.137 \pm 0.022$ |
| Sumba | $\mathbf{0.179 \pm 0.000}$ | $\underline{0.286 \pm 0.001}$ | $\mathbf{0.850 \pm 0.001}$ | $\mathbf{0.192 \pm 0.001}$ | $0.521 \pm 0.002$ | $\mathbf{2.042 \pm 0.015}$ |

| Method | Electricity | | | Weather | | |
|---|---|---|---|---|---|---|
| (F = 6) | MAE | RMSE | MAPE(%) | MAE | RMSE | MAPE(%) |
| MICN [33] | $0.235 \pm 0.002$ | $0.317 \pm 0.001$ | $1.890 \pm 0.009$ | $0.111 \pm 0.001$ | $0.250 \pm 0.001$ | $2.886 \pm 0.013$ |
| ModernTCN [29] | $0.191 \pm 0.002$ | $0.286 \pm 0.001$ | $1.719 \pm 0.005$ | $0.0934 \pm 0.000$ | $0.241 \pm 0.001$ | $\underline{1.712 \pm 0.020}$ |
| PatchTST [30] | $0.188 \pm 0.001$ | $0.290 \pm 0.001$ | $1.547 \pm 0.009$ | $\underline{0.0746 \pm 0.000}$ | $0.239 \pm 0.001$ | $3.138 \pm 0.096$ |
| FEDformer [45] | $0.271 \pm 0.002$ | $0.384 \pm 0.001$ | $2.541 \pm 0.008$ | $0.230 \pm 0.001$ | $0.364 \pm 0.003$ | $4.118 \pm 0.027$ |
| Autoformer [35] | $0.269 \pm 0.004$ | $0.378 \pm 0.002$ | $2.643 \pm 0.007$ | $0.246 \pm 0.002$ | $0.399 \pm 0.003$ | $12.782 \pm 1.039$ |
| Reformer [19] | $0.348 \pm 0.001$ | $0.483 \pm 0.001$ | $3.189 \pm 0.006$ | $0.117 \pm 0.001$ | $0.263 \pm 0.002$ | $3.611 \pm 0.025$ |
| S4 [12] | $0.341 \pm 0.003$ | $0.489 \pm 0.002$ | $3.265 \pm 0.009$ | $0.0926 \pm 0.001$ | $0.285 \pm 0.003$ | $3.932 \pm 0.037$ |
| MTGNN [38] | $0.169 \pm 0.001$ | $0.268 \pm 0.002$ | $1.636 \pm 0.003$ | $0.0817 \pm 0.000$ | $0.256 \pm 0.001$ | $1.987 \pm 0.017$ |
| MegaCRN [15] | $0.292 \pm 0.002$ | $0.419 \pm 0.002$ | $2.743 \pm 0.011$ | $0.0819 \pm 0.000$ | $0.242 \pm 0.001$ | $1.972 \pm 0.019$ |
| iTransformer [28] | $0.171 \pm 0.002$ | $0.269 \pm 0.001$ | $1.578 \pm 0.005$ | $0.0804 \pm 0.000$ | $0.243 \pm 0.002$ | $1.774 \pm 0.023$ |
| CrossFormer [42] | $0.169 \pm 0.001$ | $0.270 \pm 0.001$ | $1.541 \pm 0.006$ | $0.0882 \pm 0.000$ | $\underline{0.237 \pm 0.002}$ | $2.931 \pm 0.059$ |
| Card [34] | $\underline{0.166 \pm 0.001}$ | $\underline{0.267 \pm 0.001}$ | $1.576 \pm 0.002$ | $0.0778 \pm 0.000$ | $0.242 \pm 0.002$ | $1.748 \pm 0.009$ |
| ESG [39] | $0.168 \pm 0.002$ | $0.275 \pm 0.003$ | $1.549 \pm 0.003$ | $0.0932 \pm 0.005$ | $0.242 \pm 0.001$ | $2.322 \pm 0.026$ |
| FourierGNN [40] | $0.237 \pm 0.003$ | $0.369 \pm 0.002$ | $1.978 \pm 0.007$ | $0.0935 \pm 0.004$ | $0.257 \pm 0.002$ | $2.869 \pm 0.038$ |
| TPGNN [26] | $0.234 \pm 0.002$ | $0.367 \pm 0.002$ | $2.042 \pm 0.007$ | $0.0782 \pm 0.000$ | $0.246 \pm 0.002$ | $1.731 \pm 0.030$ |
| Sumba | $\mathbf{0.163 \pm 0.001}$ | $\mathbf{0.260 \pm 0.001}$ | $\mathbf{1.530 \pm 0.005}$ | $\mathbf{0.0704 \pm 0.000}$ | $\mathbf{0.234 \pm 0.002}$ | $\mathbf{1.663 \pm 0.013}$ |

| Method | ETTh2 | | | Traffic | | |
|---|---|---|---|---|---|---|
| (F = 6) | MAE | RMSE | MAPE(%) | MAE | RMSE | MAPE(%) |
| MICN [33] | $0.281 \pm 0.001$ | $0.405 \pm 0.001$ | $1.692 \pm 0.006$ | $0.275 \pm 0.002$ | $0.571 \pm 0.001$ | $3.369 \pm 0.019$ |
| ModernTCN [29] | $0.272 \pm 0.002$ | $0.380 \pm 0.003$ | $1.819 \pm 0.015$ | $0.261 \pm 0.002$ | $0.597 \pm 0.001$ | $3.219 \pm 0.028$ |
| PatchTST [30] | $0.210 \pm 0.002$ | $0.328 \pm 0.002$ | $0.956 \pm 0.005$ | $0.231 \pm 0.001$ | $0.554 \pm 0.002$ | $2.465 \pm 0.012$ |
| FEDformer [45] | $0.349 \pm 0.006$ | $0.508 \pm 0.007$ | $1.523 \pm 0.021$ | $0.344 \pm 0.003$ | $0.718 \pm 0.007$ | $3.845 \pm 0.033$ |
| Autoformer [35] | $0.322 \pm 0.002$ | $0.473 \pm 0.003$ | $1.350 \pm 0.010$ | $0.360 \pm 0.002$ | $0.730 \pm 0.003$ | $3.891 \pm 0.018$ |
| Reformer [19] | $0.488 \pm 0.012$ | $0.690 \pm 0.007$ | $2.604 \pm 0.031$ | $0.338 \pm 0.002$ | $0.775 \pm 0.006$ | $3.613 \pm 0.025$ |
| S4 [12] | $0.562 \pm 0.003$ | $0.747 \pm 0.009$ | $2.755 \pm 0.021$ | $0.301 \pm 0.002$ | $0.738 \pm 0.003$ | $2.952 \pm 0.020$ |
| MTGNN [38] | $0.212 \pm 0.001$ | $\underline{0.323 \pm 0.002}$ | $0.994 \pm 0.006$ | $0.227 \pm 0.002$ | $0.551 \pm 0.004$ | $2.562 \pm 0.007$ |
| MegaCRN [15] | $0.287 \pm 0.004$ | $0.411 \pm 0.003$ | $1.437 \pm 0.009$ | $0.338 \pm 0.002$ | $0.754 \pm 0.002$ | $3.994 \pm 0.015$ |
| iTransformer [28] | $0.212 \pm 0.001$ | $0.329 \pm 0.001$ | $1.029 \pm 0.003$ | $\underline{0.211 \pm 0.002}$ | $\mathbf{0.524 \pm 0.001}$ | $2.269 \pm 0.005$ |
| CrossFormer [42] | $0.249 \pm 0.002$ | $0.387 \pm 0.001$ | $1.097 \pm 0.006$ | $0.216 \pm 0.002$ | $0.634 \pm 0.003$ | $\mathbf{2.140 \pm 0.014}$ |
| Card [34] | $\underline{0.209 \pm 0.001}$ | $\mathbf{0.319 \pm 0.001}$ | $\underline{0.951 \pm 0.004}$ | $0.215 \pm 0.001$ | $0.545 \pm 0.002$ | $2.341 \pm 0.006$ |
| ESG [39] | $0.254 \pm 0.006$ | $0.371 \pm 0.013$ | $1.027 \pm 0.012$ | $0.221 \pm 0.002$ | $0.626 \pm 0.002$ | $2.248 \pm 0.003$ |
| FourierGNN [40] | $0.265 \pm 0.002$ | $0.393 \pm 0.006$ | $1.933 \pm 0.008$ | $0.291 \pm 0.001$ | $0.665 \pm 0.002$ | $3.608 \pm 0.028$ |
| TPGNN [26] | $0.236 \pm 0.001$ | $0.379 \pm 0.003$ | $1.092 \pm 0.012$ | $0.268 \pm 0.002$ | $0.726 \pm 0.004$ | $3.185 \pm 0.035$ |
| Sumba | $\mathbf{0.204 \pm 0.001}$ | $\underline{0.323 \pm 0.001}$ | $\mathbf{0.935 \pm 0.003}$ | $\mathbf{0.210 \pm 0.002}$ | $\underline{0.529 \pm 0.002}$ | $2.196 \pm 0.013$ |

Table 2: The forecasting results with prediction horizons of 3 and 6 on PEMS and Solar datasets.

| Method | PEMS | | | Solar-Energy | | |
|---|---|---|---|---|---|---|
| ($F = 3$) | MAE | RMSE | MAPE(%) | MAE | RMSE | MAPE(%) |
| MICN [33] | $0.155 \pm 0.001$ | $0.228 \pm 0.002$ | $1.180 \pm 0.004$ | $0.077 \pm 0.000$ | $0.136 \pm 0.001$ | $0.782 \pm 0.003$ |
| ModernTCN [29] | $0.143 \pm 0.001$ | $\underline{0.211 \pm 0.000}$ | $1.102 \pm 0.003$ | $0.065 \pm 0.000$ | $0.125 \pm 0.001$ | $0.608 \pm 0.002$ |
| PatchTST [30] | $0.179 \pm 0.001$ | $0.242 \pm 0.001$ | $1.137 \pm 0.003$ | $0.082 \pm 0.000$ | $0.142 \pm 0.001$ | $0.647 \pm 0.002$ |
| FEDformer [45] | $0.246 \pm 0.002$ | $0.349 \pm 0.003$ | $1.765 \pm 0.005$ | $0.169 \pm 0.001$ | $0.247 \pm 0.001$ | $1.224 \pm 0.006$ |
| Autoformer [35] | $0.462 \pm 0.003$ | $0.670 \pm 0.002$ | $2.446 \pm 0.008$ | $0.213 \pm 0.001$ | $0.290 \pm 0.002$ | $1.340 \pm 0.004$ |
| Reformer [19] | $0.190 \pm 0.001$ | $0.300 \pm 0.002$ | $1.506 \pm 0.003$ | $0.068 \pm 0.000$ | $0.147 \pm 0.001$ | $0.818 \pm 0.004$ |
| S4 [12] | $0.232 \pm 0.002$ | $0.331 \pm 0.003$ | $2.029 \pm 0.007$ | $0.092 \pm 0.001$ | $0.139 \pm 0.001$ | $0.974 \pm 0.003$ |
| MTGNN [38] | $0.145 \pm 0.000$ | $0.215 \pm 0.000$ | $1.127 \pm 0.004$ | $0.060 \pm 0.000$ | $0.135 \pm 0.001$ | $0.629 \pm 0.002$ |
| MegaCRN [15] | $0.154 \pm 0.001$ | $0.232 \pm 0.003$ | $1.209 \pm 0.004$ | $0.057 \pm 0.000$ | $0.129 \pm 0.001$ | $0.658 \pm 0.002$ |
| iTransformer [28] | $0.146 \pm 0.001$ | $0.215 \pm 0.001$ | $1.147 \pm 0.007$ | $0.060 \pm 0.000$ | $0.126 \pm 0.001$ | $0.609 \pm 0.002$ |
| CrossFormer [42] | $0.144 \pm 0.001$ | $0.219 \pm 0.001$ | $1.139 \pm 0.002$ | $\underline{0.047 \pm 0.000}$ | $\underline{0.116 \pm 0.001}$ | $\underline{0.553 \pm 0.003}$ |
| Card [34] | $0.155 \pm 0.001$ | $0.226 \pm 0.001$ | $1.139 \pm 0.002$ | $0.057 \pm 0.000$ | $0.134 \pm 0.001$ | $0.684 \pm 0.002$ |
| ESG [39] | $\underline{0.142 \pm 0.001}$ | $0.212 \pm 0.001$ | $\underline{1.089 \pm 0.003}$ | $0.050 \pm 0.000$ | $0.122 \pm 0.001$ | $0.606 \pm 0.002$ |
| FourierGNN [40] | $0.154 \pm 0.001$ | $0.236 \pm 0.002$ | $1.292 \pm 0.003$ | $0.063 \pm 0.000$ | $0.129 \pm 0.002$ | $0.647 \pm 0.003$ |
| TPGNN [26] | $0.149 \pm 0.001$ | $0.230 \pm 0.001$ | $1.157 \pm 0.005$ | $0.059 \pm 0.000$ | $0.138 \pm 0.002$ | $0.612 \pm 0.002$ |
| Sumba | $\mathbf{0.137 \pm 0.001}$ | $\mathbf{0.204 \pm 0.000}$ | $\mathbf{1.060 \pm 0.002}$ | $\mathbf{0.046 \pm 0.000}$ | $\mathbf{0.114 \pm 0.002}$ | $\mathbf{0.541 \pm 0.002}$ |

| Method | PEMS | | | Solar-Energy | | |
|---|---|---|---|---|---|---|
| ($F = 6$) | MAE | RMSE | MAPE(%) | MAE | RMSE | MAPE(%) |
| MICN [33] | $0.176 \pm 0.001$ | $0.239 \pm 0.001$ | $1.355 \pm 0.003$ | $0.094 \pm 0.000$ | $0.175 \pm 0.001$ | $0.889 \pm 0.003$ |
| ModernTCN [29] | $0.154 \pm 0.001$ | $0.228 \pm 0.003$ | $1.218 \pm 0.006$ | $0.086 \pm 0.000$ | $0.161 \pm 0.002$ | $0.796 \pm 0.003$ |
| PatchTST [30] | $0.189 \pm 0.001$ | $0.262 \pm 0.002$ | $1.226 \pm 0.003$ | $0.096 \pm 0.000$ | $0.183 \pm 0.001$ | $0.841 \pm 0.002$ |
| FEDformer [45] | $0.244 \pm 0.001$ | $0.348 \pm 0.002$ | $1.761 \pm 0.002$ | $0.169 \pm 0.001$ | $0.253 \pm 0.001$ | $1.305 \pm 0.002$ |
| Autoformer [35] | $0.515 \pm 0.002$ | $0.729 \pm 0.004$ | $2.881 \pm 0.003$ | $0.246 \pm 0.002$ | $0.333 \pm 0.002$ | $1.528 \pm 0.002$ |
| Reformer [19] | $0.202 \pm 0.001$ | $0.316 \pm 0.002$ | $1.655 \pm 0.003$ | $0.083 \pm 0.000$ | $0.175 \pm 0.001$ | $0.919 \pm 0.004$ |
| S4 [12] | $0.244 \pm 0.003$ | $0.344 \pm 0.005$ | $2.177 \pm 0.006$ | $0.101 \pm 0.001$ | $0.158 \pm 0.002$ | $1.156 \pm 0.004$ |
| MTGNN [38] | $0.158 \pm 0.001$ | $0.234 \pm 0.001$ | $1.229 \pm 0.003$ | $0.083 \pm 0.000$ | $0.169 \pm 0.001$ | $0.810 \pm 0.002$ |
| MegaCRN [15] | $0.167 \pm 0.002$ | $0.254 \pm 0.003$ | $1.356 \pm 0.005$ | $0.087 \pm 0.000$ | $0.172 \pm 0.002$ | $0.924 \pm 0.005$ |
| iTransformer [28] | $0.157 \pm 0.001$ | $0.233 \pm 0.002$ | $1.241 \pm 0.003$ | $0.071 \pm 0.000$ | $0.163 \pm 0.001$ | $0.823 \pm 0.002$ |
| CrossFormer [42] | $\underline{0.151 \pm 0.001}$ | $0.233 \pm 0.002$ | $1.234 \pm 0.002$ | $0.063 \pm 0.001$ | $\underline{0.156 \pm 0.002}$ | $\underline{0.739 \pm 0.003}$ |
| Card [34] | $0.152 \pm 0.002$ | $0.228 \pm 0.003$ | $1.182 \pm 0.004$ | $\underline{0.062 \pm 0.000}$ | $0.160 \pm 0.002$ | $0.775 \pm 0.003$ |
| ESG [39] | $0.151 \pm 0.001$ | $\underline{0.226 \pm 0.001}$ | $1.159 \pm 0.004$ | $0.068 \pm 0.000$ | $0.158 \pm 0.001$ | $0.757 \pm 0.005$ |
| FourierGNN [40] | $0.182 \pm 0.003$ | $0.267 \pm 0.002$ | $1.327 \pm 0.003$ | $0.096 \pm 0.001$ | $0.177 \pm 0.002$ | $0.872 \pm 0.002$ |
| TPGNN [26] | $0.169 \pm 0.001$ | $0.256 \pm 0.002$ | $1.292 \pm 0.003$ | $0.090 \pm 0.000$ | $0.196 \pm 0.002$ | $0.872 \pm 0.002$ |
| Sumba | $\mathbf{0.147 \pm 0.001}$ | $\mathbf{0.222 \pm 0.002}$ | $\mathbf{1.155 \pm 0.002}$ | $\mathbf{0.062 \pm 0.000}$ | $\mathbf{0.147 \pm 0.001}$ | $\mathbf{0.734 \pm 0.002}$ |

methods, and the history window size $H$ is set to 168. All methods are trained on Nvidia V100 GPUs. Our method is implemented with PyTorch 2.0 and we use the source codes publicly released by the authors for all baseline methods. We adjust the hyperparameters of baseline methods to obtain the best performance on each dataset, and evaluate the performance of different methods in terms of Mean Absolute Error (MAE), Root Mean Square Error (RMSE), and Mean Absolute Percentage Error (MAPE).

## 4.2 Forecasting Performance

Table 1 and 2 present the forecasting performance with $F \in \{3, 6\}$ on six benchmark datasets, with results evaluated using three different random seeds. The best results are highlighted in **bold**, while the second-best results are underlined. PatchTST and ModernTCN demonstrate the best performance among methods without explicitly modeling spatial correlations. Besides, graph-based spatio-temporal methods yield better results than those that do not account for spatial structures. In addition, Card, Crossformer, iTransformer, and ESG outperform static graph-based methods such as MTGNN and MegaCRN, highlighting the significance of explicitly modeling dynamic spatial structures. Our method Sumba achieves state-of-the-art performance in most cases, demonstrating an improvement up to $8.5\%$ over the best baseline on the Weather dataset. This superior performance is attributed to its capability to produce dynamic graph structures with low variance and high expressiveness using the proposed structured matrix basis. The forecasting results for additional prediction horizons are provided in Appendix C.

Table 3: Ablation study.

| $F=3$ | Electricity | | | Weather | | | PEMS | | |
|---|---|---|---|---|---|---|---|---|---|
| | MAE | RMSE | MAPE(%) | MAE | RMSE | MAPE(%) | MAE | RMSE | MAPE(%) |
| w/o. dynamic | 0.153 | 0.245 | 1.340 | 0.0693 | 0.219 | 1.624 | 0.145 | 0.215 | 1.127 |
| w/. $\mathbf{U}_m, \mathbf{V}_m$ | 0.148 | 0.238 | 1.251 | 0.0659 | 0.214 | 2.255 | 0.138 | 0.205 | 1.054 |
| w/o. orthogonality | 0.151 | 0.242 | 1.337 | 0.0633 | 0.213 | 1.605 | 0.142 | 0.209 | 1.094 |
| Sumba | 0.148 | 0.237 | 1.331 | 0.0587 | 0.208 | 1.381 | 0.137 | 0.204 | 1.060 |
| $F=6$ | Electricity | | | Weather | | | PEMS | | |
| | MAE | RMSE | MAPE(%) | MAE | RMSE | MAPE(%) | MAE | RMSE | MAPE(%) |
| w/o. dynamic | 0.169 | 0.268 | 1.636 | 0.0817 | 0.256 | 1.987 | 0.158 | 0.234 | 1.229 |
| w/. $\mathbf{U}_m, \mathbf{V}_m$ | 0.164 | 0.262 | 1.537 | 0.0722 | 0.234 | 2.116 | 0.149 | 0.227 | 1.162 |
| w/o. orthogonality | 0.167 | 0.265 | 1.578 | 0.0706 | 0.239 | 1.690 | 0.151 | 0.226 | 1.179 |
| Sumba | 0.163 | 0.260 | 1.530 | 0.0704 | 0.234 | 1.663 | 0.147 | 0.222 | 1.155 |

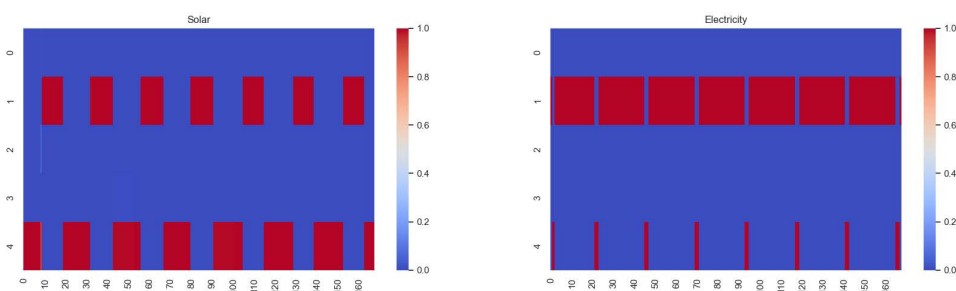

Figure 2: The change of $\boldsymbol{\alpha}$ over one week.

## 4.3 Ablation Study

We conduct the ablation study on the Electricity, Weather, and PEMS datasets with prediction horizons $F$ of 3 and 6 to verify the efficacy of the proposed modules. In particular, we consider the following variants of our proposed model: (1) **w/o. dynamic**: we replace the Dynamic GCN module with the vanilla GCN module in [38]. (2) **w/. $\mathbf{U}_m, \mathbf{V}_m$**: we use distinct $\mathbf{U}_m$, $\mathbf{V}_m$ for each basis matrix $\mathbf{B}_m$. (3) **w/o. orthogonality**: we remove the orthogonal parameterization on $\mathbf{U}$ and $\mathbf{V}$. As shown in Table 3, the performance of **w/o. dynamic** declines rapidly by up to $13.83\%$, $8.59\%$, and $16.31\%$ on three metrics, which verifies the efficacy of the proposed dynamic GCN model. The absence of structure regularization (**w/. $\mathbf{U}_m, \mathbf{V}_m$**) and the elimination of orthogonal parameterization (**w/o. orthogonality**) on structured matrix basis lead to averaged performance drops of $4.95\%$ and $3.05\%$, respectively. This proves the effectiveness of the proposed structure regularization and orthogonal constraint strategy.

## 4.4 Interpretable Dynamics

As discussed in Section 3.1, one appealing feature of our proposed Sumba is that it enables us to gain insights into the underlying time series dynamics by tracking the change of the matrix basis coefficient $\boldsymbol{\alpha}$ over time. To verify this, we present the heatmap of the change of $\boldsymbol{\alpha}$ on two datasets—Solar-Energy and Electricity in Figure 2, in which the x-axis denotes the time (hourly), the y-axis represents the no. of 5 basis matrices, and the color indicates the weight of $\boldsymbol{\alpha}$. It can be observed that most of the weights are concentrated on two basis matrices, no. 1 and no. 4, which implies that there are two dominant spatial structures on the two datasets. Furthermore, the two dominant spatial structures appear alternatively and regularly, which actually corresponds to the day and night for the Solar-Energy dataset. This aligns well with our intuition that solar energy observations should manifest different spatial correlations as the light intensity varies. For the Electricity dataset, there is a spatial correlation (no. 4) that spans two hours and only emerges in midnight, this reveals an interesting electricity consumption pattern, which is unnoticed by previous methods. Therefore, our proposed method is able to offer more interpretable dynamics of the underlying systems through tracking $\boldsymbol{\alpha}$.

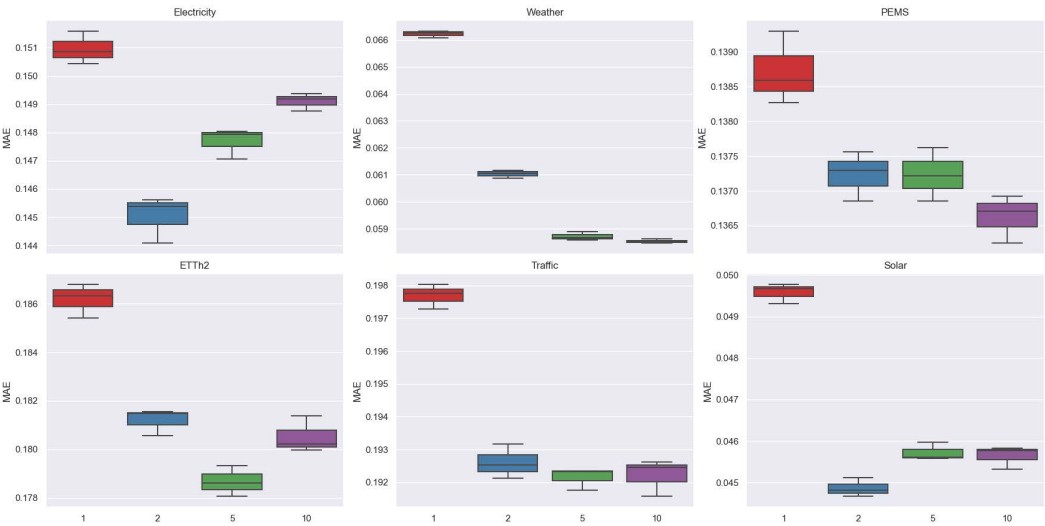

Figure 3: The sensitivity of the dimension of basis $M$.

## 4.5 Parameter Sensitivity Analysis

We evaluate the impact of $M$ on performance of our model, where the $M$ ranges from 1, 2, 5, to 10. $M = 1$ corresponds to the static graph. As shown in Figure 3, the performance of our model improves significantly when $M > 1$. When $M = 2$, our method achieves the best performance on Electricity and Solar-Energy datasets, this coincides with the fact that the two datasets have two dominant spatial structure patterns, i.e., day and night. The experiments empirically show that the best setting of $M$ is 5 on ETTh2, Traffic, and PEMS datasets and 10 on the Weather dataset. We hypothesize this is due to the Weather dataset having more complex dynamics, and thus it requires more basis matrices to cover the various patterns. The sensitivity analysis of $L$, $K$ and $H$ is presented in Appendix D.

## 5 Conclusion

In this paper, we propose a time series forecasting method with the structured matrix basis, Sumba, to capture dynamic spatial structures. To this end, we propose a novel structured parameterization and impose structure regularization on the basis to enhance parameter efficiency, the output space of the spatial structure function is thus well constrained and the generated spatial structures have lower variance. Our proposed method offers us a manner to gain insights into the dynamics of the underlying systems, and thus it is more interpretable. The experiments on six benchmark datasets verify the superiority of our proposed method. Our extensive ablation studies prove the effectiveness of each proposed component. In addition, the case study shows that our method can offer desirable interpretability. In the future, we would like to explore how to better regularize the learned matrix vector space and set the dimension of basis $M$ in a data-driven manner. We will also explore the possibility of integrating our proposed spatial structure modeling with other temporal encoders such as Transformers, and Structured State Space models, and apply our interpretable dynamics into more datasets to discover more interesting and hidden patterns.

## Acknowledgements

This work is supported by the National Natural Science Foundation of China under Grant No. 62206074, Grant No. 62306085, Grant No. 62072137, Shenzhen College Stability Support Plan under Grant No. GXWD20220811173233001, and the National Key R&D Program of China under Grant No. 2023YFB4503100.

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

# A  Background and Proof of the Theorem

## A.1  Geometry Interpretation of Matrix Multiplication

Recall that given a matrix $\mathbf{A} = \mathbf{U}\Sigma\mathbf{V}^\top$, the multiplication $\mathbf{A}\mathbf{x}$ can be interpreted as 1) transforming $\mathbf{x}$ into the new coordinate system $\mathbf{x}' \triangleq \mathbf{V}^\top\mathbf{x}$, 2) doing the elementwise product in the new coordinate $\mathbf{y}' = \Sigma\mathbf{x}'$, and 3) transforming the coordinate system again $\mathbf{y} = \mathbf{U}\mathbf{y}'$.

$$\mathbf{A}\mathbf{x} = \mathbf{U}\Sigma\mathbf{V}^\top\mathbf{x} = \mathbf{U}\Sigma\mathbf{x}' = \mathbf{U}\mathbf{y}'. \tag{10}$$

## A.2  Proof of the Theorem

**Theorem A.1.** *The output space of $f_{\mathrm{spatial}}$ in Eq. 4 is bounded by the sum of the maximum of $\Sigma_m$(the maximum singular value of $\mathbf{B}_m$) for $m = 1, 2, \ldots, M$ in terms of the $\ell_2$ norm i.e., $\|f_{\mathrm{spatial}}\|_2 \leq \sum_{m=1}^{M} \max(\Sigma_m)$.*

*Proof.*

$$\|f_{\mathrm{spatial}}\|_2 = \left\| \sum_{m=1}^{M} \alpha_m \mathbf{U}\Sigma_m\mathbf{V}^\top \right\|_2 \tag{11}$$

$$\leq \sum_{m=1}^{M} \alpha_m \left\| \mathbf{U}\Sigma_m\mathbf{V}^\top \right\|_2 \quad \text{(triangle inequality)} \tag{12}$$

$$= \sum_{m=1}^{M} \left\| \mathbf{U}\Sigma_m\mathbf{V}^\top \right\|_2 \quad\quad (0 < \alpha_m < 1) \tag{13}$$

$$= \sum_{m=1}^{M} \|\mathbf{U}\|_2 \|\Sigma_m\|_2 \|\mathbf{V}^\top\|_2 \tag{14}$$

$$= \sum_{m=1}^{M} \|\Sigma_m\|_2 \quad\quad (\mathbf{U}, \mathbf{V} \text{ are orthogonal matrices}) \tag{15}$$

$$= \sum_{m=1}^{M} \max(\Sigma_m). \tag{16}$$

The proof is completed. $\square$

# B  Implementation Details

## B.1  Baselines

In our experiments, we compare our proposed Sumba with state-of-the-art (SOTA) baseline models.

- MICN: It utilizes temporal convolution networks and isometric convolution networks to capture local and global temporal correlations, respectively.
- ModernTCN: It leverages large kernels to build long-term temporal dependencies.
- PatchTST: It introduces a patch-based attention mechanism and channel-independent strategy to establish long-term temporal correlations.
- FEDformer: It leverages frequency-enhanced attention to capture long-term temporal dependencies and employs frequency sampling to reduce the complexity.
- Autoformer: It designs auto-correlation module to discover the sub-series similarity based on periodicity and aggregates similar sub-series from underlying periods.
- Reformer: It utilizes locality sensitive hashing strategy to reduce the complexity of the attention mechanism.
- S4: It captures long-term temporal dependencies via structured transition matrices.

- MTGNN: It builds an adaptive directed graph using learnable node embedding and aggregates information along spatial dimensions through mix-hop propagation.

- MegaCRN: It exploits memory-enhanced node embedding to build the graph structure.

- iTransformer: It embeds the whole time series into a spatial token and captures spatial correlations using the self-attention mechanism.

- Crossformer: It constructs both temporal and spatial dependencies using the attention mechanism and introduces a router mechanism to decrease the complexity of the spatial attention module.

- Card: It introduces summarized spatial tokens to decrease the complexity of the spatial attention module and employs a token blend mechanism in the temporal attention module to extract local temporal correlations.

- ESG: It designs multi-scale evolving node embedding based on Gated Recurrent Unit to capture dynamic spatial dependencies.

- FourierGNN: It builds dynamic graphs by merging spatial and temporal dimensions and designs the Fourier Graph Operator to decrease complexity.

- TPGNN: It regards the dynamic spatial correlations as the matrix polynomial graph where the time-varying coefficients are determined by timestamps.

## B.2 Metrics

The evaluation metrics including MAE, RMSE, and MAPE are as follows:

$$\text{MAE} = \frac{\sum_{ij \in \Omega} |y_{ij} - \hat{y}_{ij}|}{|\Omega|} \quad \text{RMSE} = \sqrt{\frac{\sum_{ij \in \Omega} (y_{ij} - \hat{y}_{ij})^2}{|\Omega|}} \quad \text{MAPE} = \sum_{ij \in \Omega} \frac{|y_{ij} - \hat{y}_{ij}|}{|\Omega| \cdot |y_{ij}|} \quad (17)$$

where $\Omega$ denotes the index set along temporal and spatial dimensions.

## C  More Forecasting Results

Table 4 provides the results with prediction lengths of 12 and 24. The results underscore the significant role of explicitly capturing spatial correlations in improving forecasting accuracy. Furthermore, dynamic graphs exhibit a greater ability to enhance the forecasting performance than modeling static spatial correlations. Among the baseline methods, Card, Crossformer, and ESG demonstrate superior forecasting accuracy. Notably, our model achieves state-of-the-art performance, yielding improvements more than $20\%$ in terms of MAE and $25\%$ in terms of MAPE over the existing methods on the Solar-Energy dataset. In addition, we conduct experiments with a forecasting horizon of 96, and as Table 5 shows, Sumba gives rise to favorable performance in comparison to PatchTST, FEDformer, and iTransformer which are designed specifically for long-term forecasting.

## D  Hyperparameters Sensitivity

We evaluate the sensitivity of hyperparameters including the number of blocks $L$, the rank $K$, and the history window size $H$. **Number of Blocks**: As shown in Figure 4, Sumba achieves its best performance when $L = 3$. Further increasing $L$ does not lead to performance improvements. We hypothesize that this is due to the over-smoothing issue associated with Graph Neural Networks. **Rank**: Figure 5 shows how performance varies as the rank $K$ changes from 20 to 50. The optimal setting for $K$ is 30 for the Electricity, PEMS, and Solar-Energy datasets, while $K = 50$ yields the best results for the Traffic dataset. We propose that this is attributed to the larger number of nodes in the Traffic dataset, which requires a higher rank to effectively represent the dynamic spatial correlations. **History window**: Figure 6 demonstrates that increasing the history window size enhances model performance, especially for the Electricity dataset.

Table 4: The forecasting results under forecasting length 12 and 24.

| Method (F = 12) | Electricity MAE | RMSE | MAPE(%) | Weather MAE | RMSE | MAPE(%) | PEMS MAE | RMSE | MAPE(%) |
|---|---|---|---|---|---|---|---|---|---|
| MICN [33] | 0.278 | 0.390 | 2.633 | 0.123 | 0.276 | 2.970 | 0.277 | 0.404 | 2.021 |
| ModernTCN [29] | 0.202 | 0.307 | 1.796 | 0.111 | 0.268 | 4.541 | 0.169 | 0.252 | 1.368 |
| PatchTST [30] | 0.201 | 0.311 | 1.765 | 0.0958 | 0.270 | 6.108 | 0.211 | 0.293 | 1.426 |
| FEDformer [45] | 0.274 | 0.390 | 2.535 | 0.251 | 0.394 | 6.789 | 0.262 | 0.369 | 1.931 |
| Autoformer [35] | 0.284 | 0.402 | 2.822 | 0.216 | 0.368 | 8.119 | 0.585 | 0.732 | 2.736 |
| Reformer [19] | 0.376 | 0.521 | 3.495 | 0.185 | 0.349 | 4.815 | 0.217 | 0.338 | 1.801 |
| S4 [12] | 0.361 | 0.512 | 3.314 | 0.213 | 0.347 | 5.443 | 0.246 | 0.346 | 2.184 |
| MTGNN [38] | 0.184 | 0.292 | 1.704 | 0.0899 | 0.275 | 4.549 | 0.177 | 0.263 | 1.391 |
| MegaCRN [15] | 0.366 | 0.520 | 2.715 | 0.115 | 0.279 | 3.757 | 0.188 | 0.286 | 1.572 |
| iTransformer [28] | 0.188 | 0.296 | 1.703 | 0.0978 | 0.273 | 4.879 | 0.173 | 0.258 | 1.402 |
| CrossFormer [42] | 0.183 | 0.289 | 1.715 | 0.122 | 0.270 | 4.656 | 0.164 | 0.256 | 1.370 |
| Card [34] | 0.186 | 0.285 | 1.610 | 0.0892 | 0.268 | 3.829 | 0.168 | 0.254 | 1.347 |
| ESG [39] | 0.189 | 0.295 | 1.691 | 0.0969 | 0.264 | 3.513 | 0.164 | 0.248 | 1.313 |
| FourierGNN [40] | 0.275 | 0.396 | 2.694 | 0.121 | 0.276 | 4.932 | 0.199 | 0.305 | 1.832 |
| TPGNN [26] | 0.259 | 0.308 | 2.165 | 0.104 | 0.282 | 2.898 | 0.186 | 0.287 | 1.577 |
| Sumba | **0.178** | **0.283** | **1.560** | **0.0884** | **0.263** | 3.794 | **0.162** | **0.245** | **1.295** |

| Method (F = 12) | ETTh2 MAE | RMSE | MAPE(%) | Traffic MAE | RMSE | MAPE(%) | Solar-Energy MAE | RMSE | MAPE(%) |
|---|---|---|---|---|---|---|---|---|---|
| MICN [33] | 0.290 | 0.414 | 1.492 | 0.281 | 0.629 | 3.479 | 0.105 | 0.207 | 1.000 |
| ModernTCN [29] | 0.320 | 0.445 | 1.564 | 0.271 | 0.612 | 3.219 | 0.118 | 0.210 | 0.974 |
| PatchTST [30] | 0.237 | 0.374 | 1.135 | 0.250 | 0.582 | 2.559 | 0.122 | 0.243 | 1.054 |
| FEDformer [45] | 0.349 | 0.512 | 1.550 | 0.350 | 0.727 | 3.840 | 0.204 | 0.292 | 1.289 |
| Autoformer [35] | 0.360 | 0.526 | 1.481 | 0.361 | 0.735 | 4.007 | 0.309 | 0.414 | 1.680 |
| Reformer [19] | 0.579 | 0.728 | 4.289 | 0.354 | 0.789 | 3.608 | 0.113 | 0.229 | 1.125 |
| S4 [12] | 0.634 | 0.846 | 2.768 | 0.307 | 0.749 | 3.079 | 0.137 | 0.210 | 1.585 |
| MTGNN [38] | 0.245 | 0.381 | 1.172 | 0.247 | 0.577 | 2.762 | 0.0975 | 0.214 | 1.007 |
| MegaCRN [15] | 0.332 | 0.465 | 1.519 | 0.367 | 0.784 | 4.397 | 0.116 | 0.226 | 1.088 |
| iTransformer [28] | 0.242 | 0.376 | 1.121 | 0.237 | **0.559** | 2.582 | 0.0974 | 0.211 | 0.991 |
| CrossFormer [42] | 0.307 | 0.446 | 1.467 | 0.238 | 0.666 | 2.497 | 0.0881 | 0.197 | 0.936 |
| Card [34] | 0.238 | 0.378 | 1.076 | 0.228 | 0.576 | 2.455 | 0.0850 | 0.210 | 0.996 |
| ESG [39] | 0.313 | 0.454 | 1.179 | 0.269 | 0.667 | 2.868 | 0.0935 | 0.204 | 0.962 |
| FourierGNN [40] | 0.309 | 0.454 | 1.551 | 0.382 | 0.792 | 4.159 | 0.119 | 0.208 | 1.098 |
| TPGNN [26] | 0.269 | 0.427 | 1.283 | 0.302 | 0.780 | 2.507 | 0.107 | 0.205 | 0.976 |
| Sumba | **0.234** | **0.369** | **1.045** | **0.225** | 0.569 | **2.358** | **0.0845** | **0.191** | **0.897** |

| Method (F = 24) | Electricity MAE | RMSE | MAPE(%) | Weather MAE | RMSE | MAPE(%) | PEMS MAE | RMSE | MAPE(%) |
|---|---|---|---|---|---|---|---|---|---|
| MICN [33] | 0.261 | 0.375 | 2.595 | 0.149 | 0.309 | **5.026** | 0.197 | 0.292 | 1.646 |
| ModernTCN [29] | 0.222 | 0.331 | 1.945 | 0.151 | 0.306 | 7.100 | 0.197 | 0.290 | 1.582 |
| PatchTST [30] | 0.211 | 0.329 | 1.859 | 0.128 | 0.309 | 10.480 | 0.246 | 0.345 | 1.611 |
| FEDformer [45] | 0.281 | 0.400 | 2.782 | 0.254 | 0.407 | 10.152 | 0.282 | 0.394 | 1.983 |
| Autoformer [35] | 0.278 | 0.398 | 2.932 | 0.234 | 0.389 | 9.728 | 0.455 | 0.601 | 2.277 |
| Reformer [19] | 0.402 | 0.554 | 3.829 | 0.248 | 0.422 | 8.590 | 0.263 | 0.400 | 2.435 |
| S4 [12] | 0.348 | 0.499 | 3.258 | 0.269 | 0.429 | 8.225 | 0.259 | 0.361 | 2.403 |
| MTGNN [38] | 0.194 | 0.311 | 1.777 | 0.119 | 0.331 | 8.730 | 0.209 | 0.307 | 1.617 |
| MegaCRN [15] | 0.337 | 0.483 | 2.593 | 0.144 | 0.311 | 7.949 | 0.230 | 0.343 | 1.937 |
| iTransformer [28] | 0.195 | 0.310 | 1.774 | 0.126 | 0.313 | 8.254 | 0.200 | 0.296 | 1.552 |
| CrossFormer [42] | 0.196 | 0.310 | 1.944 | 0.160 | 0.309 | 7.186 | 0.189 | 0.283 | 1.565 |
| Card [34] | 0.197 | 0.315 | 1.746 | **0.115** | 0.303 | 8.007 | 0.196 | 0.294 | 1.528 |
| ESG [39] | 0.196 | 0.314 | 1.729 | 0.127 | **0.294** | 5.227 | **0.182** | 0.296 | 1.507 |
| FourierGNN [40] | 0.316 | 0.396 | 2.409 | 0.133 | 0.259 | 11.606 | 0.245 | 0.367 | 1.518 |
| TPGNN [26] | 0.262 | 0.355 | 2.121 | 0.125 | 0.296 | 5.238 | 0.226 | 0.319 | 1.903 |
| Sumba | **0.189** | **0.303** | **1.696** | **0.115** | 0.295 | 8.593 | 0.186 | **0.281** | **1.488** |

| Method (F = 24) | ETTh2 MAE | RMSE | MAPE(%) | Traffic MAE | RMSE | MAPE(%) | Solar-Energy MAE | RMSE | MAPE(%) |
|---|---|---|---|---|---|---|---|---|---|
| MICN [33] | 0.313 | 0.464 | 1.345 | 0.294 | 0.693 | 3.107 | 0.146 | 0.284 | 1.216 |
| ModernTCN [29] | 0.479 | 0.631 | 3.095 | 0.281 | 0.626 | 3.261 | 0.160 | 0.279 | 1.176 |
| PatchTST [30] | 0.281 | 0.440 | 1.233 | 0.258 | 0.594 | 2.613 | 0.166 | 0.310 | 1.219 |
| FEDformer [45] | 0.368 | 0.542 | 1.651 | 0.353 | 0.740 | 4.142 | 0.229 | 0.345 | 1.533 |
| Autoformer [35] | 0.398 | 0.579 | 1.690 | 0.354 | 0.743 | 3.519 | 0.355 | 0.502 | 1.929 |
| Reformer [19] | 0.754 | 0.955 | 5.485 | 0.378 | 0.815 | 3.982 | 0.189 | 0.328 | 1.585 |
| S4 [12] | 0.6378 | 0.841 | 2.844 | 0.313 | 0.754 | 3.291 | 0.174 | 0.272 | 2.124 |
| MTGNN [38] | 0.288 | 0.452 | 1.267 | 0.266 | 0.602 | 3.117 | 0.138 | 0.286 | 1.224 |
| MegaCRN [15] | 0.363 | 0.516 | 1.738 | 0.415 | 0.833 | 4.990 | 0.151 | 0.296 | 1.508 |
| iTransformer [28] | 0.276 | 0.436 | 1.251 | 0.247 | **0.577** | 2.762 | 0.137 | 0.290 | 1.210 |
| CrossFormer [42] | 0.407 | 0.588 | 1.774 | 0.248 | 0.676 | 2.675 | **0.121** | 0.278 | 1.121 |
| Card [34] | 0.274 | 0.439 | **1.196** | 0.247 | 0.578 | **2.481** | 0.122 | 0.294 | 1.278 |
| ESG [39] | 0.308 | 0.457 | 1.271 | 0.275 | 0.689 | 2.715 | 0.124 | 0.271 | 1.166 |
| FourierGNN [40] | 0.326 | 0.515 | 1.374 | 0.427 | 0.794 | 4.189 | 0.162 | 0.307 | 1.571 |
| TPGNN [26] | 0.293 | 0.445 | 1.271 | 0.349 | 0.728 | 3.319 | 0.140 | 0.277 | 1.253 |
| Sumba | **0.273** | **0.433** | 1.217 | **0.243** | 0.589 | 2.669 | **0.121** | **0.264** | **1.119** |

Table 5: The results with forecasting horizon 96.

| Method | Electricity | | | Weather | | | PEMS | | |
|---|---|---|---|---|---|---|---|---|---|
| | MAE | RMSE | MAPE (%) | MAE | RMSE | MAPE (%) | MAE | RMSE | MAPE (%) |
| PatchTST [30] | 0.241 | 0.378 | 2.383 | 0.206 | 0.404 | 13.692 | 0.354 | 0.497 | 1.906 |
| FEDformer [45] | 0.308 | 0.438 | 3.223 | 0.346 | 0.525 | 8.949 | 0.482 | 0.631 | 3.182 |
| Autoformer [35] | 0.303 | 0.433 | 2.923 | 0.339 | 0.518 | 21.389 | 0.721 | 0.927 | 3.447 |
| Reformer [19] | 0.389 | 0.549 | 3.675 | 0.470 | 0.695 | 11.815 | 0.403 | 0.534 | 3.151 |
| iTransformer [28] | 0.229 | 0.367 | **2.248** | 0.213 | 0.411 | 13.009 | 0.311 | 0.446 | 2.055 |
| Crossformer [42] | 0.232 | 0.366 | 2.465 | 0.211 | 0.384 | **7.745** | 0.271 | 0.405 | 2.421 |
| Card [34] | **0.225** | **0.364** | 2.310 | 0.191 | 0.389 | 9.293 | 0.283 | 0.414 | 1.871 |
| Sumba | 0.234 | 0.373 | 2.675 | **0.185** | **0.383** | 11.561 | **0.253** | **0.377** | **1.700** |

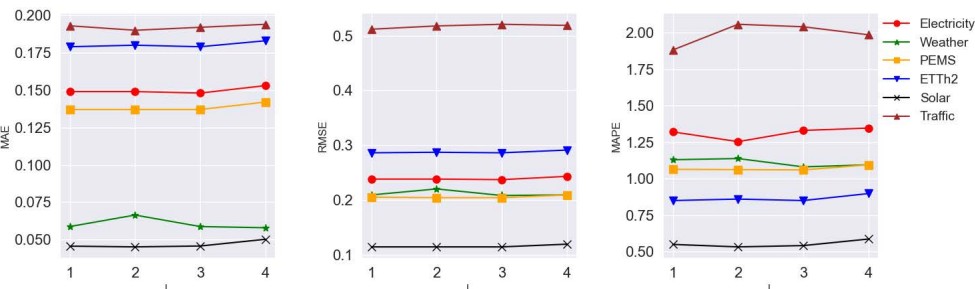

Figure 4: The sensitivity of our method to the number of blocks $L$.

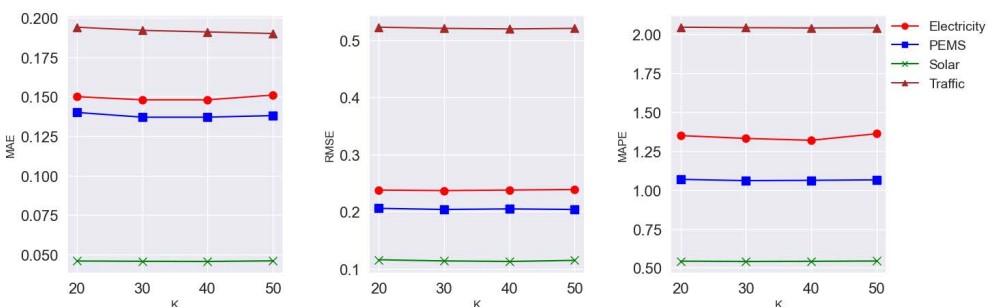

Figure 5: The sensitivity of our method to rank $K$.

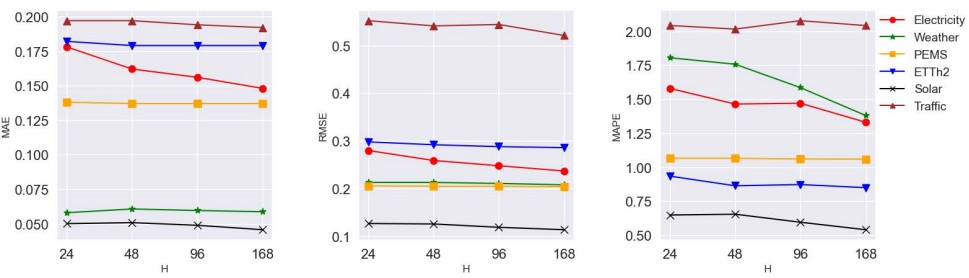

Figure 6: The sensitivity of our method to the history window $H$.

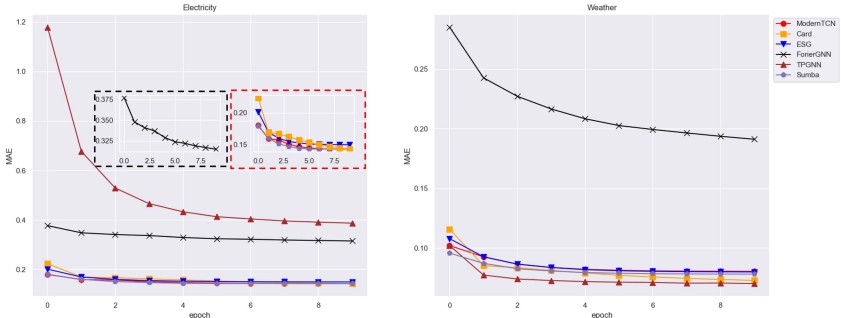

Figure 7: The training curves on Electricity and Weather datasets with prediction length $F = 3$.

Table 6: The complexity of different methods.

| Method | Complexity |
|---|---|
| MTGNN [38] | $\mathcal{O}(N^2 D)$ |
| MegaCRN [15] | $\mathcal{O}(N^2 D)$ |
| iTransformer [28] | $\mathcal{O}(N^2 D)$ |
| CrossFormer [42] | $\mathcal{O}(NrD)$ |
| Card [34] | $\mathcal{O}(NrD)$ |
| ESG [39] | $\mathcal{O}(N^2 D)$ |
| FourierGNN [40] | $\mathcal{O}(NH\log(NH))$ |
| TPGNN [26] | $\mathcal{O}(N^2 D)$ |
| w/o. Common $\mathbf{U}$-$\mathbf{V}$ basis | $\mathcal{O}(MNKD)$ |
| Sumba | $\mathcal{O}(NKD)$ |

## E  Convergence Speed and Computational Complexity

**Convergence Speed**  Figure 7 illustrates the training curves for the first 10 epochs on the Electricity and Weather datasets. Sumba, ModernTCN, and ESG exhibit fast convergence, achieving it within 5 epochs. In contrast, methods such as Card, FourierGNN, and TPGNN take several tens of epochs to reach convergence.

**Computational Complexity**  As shown in Table 6, we analyze the computational complexity of different methods in capturing spatial dependencies, where $N$ is the number of nodes, $M$ is the dimension of the matrix basis, $D$ is the dimension of node embedding or hidden representation, $r \ll N$ and $K \ll N$. MTGNN, MegaCRN, iTransformer, ESG, and TPGNN build the graph structure via inner product of static or dynamic node embedding, leading to a computation complexity of $\mathcal{O}(N^2 D)$. Crossformer and Card employ router mechanism and summarized token, respectively, in spatial correlation modeling, reducing the complexity to $\mathcal{O}(Nrd)$. FourierGNN has complexity of $\mathcal{O}(NH\log(NH))$ due to building a hypervariable graph. Our proposed Sumba leverages low-rank approximation and common coordinate transformations strategy, achieving the complexity of $\mathcal{O}(KND)$ whereas the computational cost for the model without a common $\mathbf{U}$-$\mathbf{V}$ basis is $\mathcal{O}(MKND)$. Thus, our model has low complexity, especially when $N$ is large.

## F  Limitations

Although our proposed Sumba demonstrates superior performance in multivariate time series forecasting, the dimension of matrix basis $M$ is empirically tuned in our experiments. Besides, the long-term forecasting capability of the proposed model requires further enhancement.

## G  Broader Impact

In this paper, we propose a novel time series forecasting method to capture dynamic spatial correlations with a structured matrix basis. Our research aims to contribute to the advancement of the relevant community without any negative social impact.

