# OpenReview forum: "Structured Matrix Basis for Multivariate Time Series Forecasting with Interpretable Dynamics"
_NeurIPS.cc/2024/Conference — NeurIPS 2024 poster_

### Official Review · Reviewer_VUH9 · 2024-07-08

**Soundness:** 3
**Presentation:** 3
**Contribution:** 3
**Rating:** 7
**Confidence:** 3

**Summary:**

This paper proposes a novel approach for effectively capturing spatial and temporal correlations in multivariate time series forecasting and enhancing the interpretability of prediction models. The aim is to address the shortcomings of existing methods, which often fail to adequately reflect dynamic spatial correlations and exhibit high variability. To tackle this issue, the paper introduces a method that bypasses the traditional two-stage learning process and directly generates dynamic structures using a structured matrix basis. Furthermore, the basis matrices are parameterized through singular value decomposition (SVD), and all basis matrices share the same orthogonal matrices to improve the efficiency of model training.

**Strengths:**

1. The method of bypassing the two-stage learning process and directly generating dynamic spatial structures effectively overcomes the limitations of existing models.
2. By providing interpretability of the model, it offers users greater insights and increases the reliability of the results. This is a crucial aspect that is often required in many time series forecasting research papers.
3. This paper effectively addresses various limitations that arise in traditional time series forecasting problems.
4. This paper is well-organized theoretically.

**Weaknesses:**

1. The proposed model may be complex to implement due to the use of structured matrix bases and singular value decomposition, potentially leading to a steep initial learning curve in practical applications. Does the appendix indicate similar results for datasets other than Electricity?
2. While the model has been validated on various datasets, it remains to be seen how well it performs on very specialized domain data or data with unclear characteristics (i.e., weak temporal dependencies or spatial correlations). How does the model address these scenarios?

**Questions:**

1. The proposed model may be complex to implement due to the use of structured matrix bases and singular value decomposition, potentially leading to a steep initial learning curve in practical applications. Does the appendix indicate similar results for datasets other than Electricity?
2. While the model has been validated on various datasets, it remains to be seen how well it performs on very specialized domain data or data with unclear characteristics (i.e., weak temporal dependencies or spatial correlations). How does the model address these scenarios?

**Limitations:**

Including more analysis and comparison with graph-based spatio-temporal models would enhance the paper.

---

> ### Author Rebuttal · Authors · 2024-08-04
>
> >Q1 The proposed model may be complex to implement due to the use of structured matrix bases and singular value decomposition, potentially leading to a steep initial learning curve in practical applications. Does the appendix indicate similar results for datasets other than Electricity?
>
>
> We would like to clarify that our method actually is very easy to implement. Note that the **singular value decomposition only serves to offer a parameterization form, we do not need to implement it in practice**. Instead, we only need to implement the parameterization of two orthogonal matrices $\mathbf{U}$ and $\mathbf{V}$ via the Clay map. As we mentioned in the paper, the Clay map can be implemented in a numerically stable way **in PyTorch with one line code by the  `torch.linalg.solve` function**. The learning curves of our proposed method are also very stable across all six datasets, and we will add more of them to the revised manuscript.
>
>
>
> >Q2 While the model has been validated on various datasets, it remains to be seen how well it performs on very specialized domain data or data with unclear characteristics (i.e., weak temporal dependencies or spatial correlations). How does the model address these scenarios?
>
>
> Thanks for your comments. To our knowledge, the temporal dependency assumption is necessary for all time series forecasting methods. If a dataset has only **weak temporal dependencies**, it will impede the performance of all existing time series forecasting methods. For datasets with **weak spatial correlation**, it means that different channels (series) are independent and our proposed method will consequently learn an identity matrix for the spatial structure. This will not cause any negative impact on the forecasting performance. We will add the corresponding experiments to verify this by synthesizing the datasets with independent channels in the revised manuscript.
>
>
> >Q3 Including more analysis and comparison with graph-based spatio-temporal models would enhance the paper.
>
> Thanks for your suggestion. In Table 1 of our manuscript, MTGNN, MegaCRN, iTransformer, Crossformer, Card, ESG, FourierGNN, and TPGNN are all GNN-based methods. First, we note that the graph-based spatio-temporal methods often produce better results than the methods without considering the spatial structures. Second, the dynamic graph-based methods such as iTransformer, Card, and ESG often give rise to better performance than the static graph-based methods such as MTGNN and MegaCRN. However, the existing graph-based spatio-temporal models all rely on a two-stage spatial structure learning process that is prone to yielding high variance and impeding the final performance, which is verified by the experiments. We will add more comparisons and analyses in the next version of our paper.

---

> > ### Comment · Reviewer_VUH9 · 2024-08-12
> >
> > Thanks for your response, after reading your response and other reviews, I am updating my score.

---

> > > ### Author Response · Authors · 2024-08-12
> > >
> > > Thank you for your timely response and the suggestions for improving this paper. We will further revise the paper according to your suggestions and the results in the rebuttal, including adding more learning curves, the corresponding experiments to verify the performance on datasets with independent channels, and more analyses and comparisons with graph-based spatio-temporal models.

---

### Official Review · Reviewer_aCCB · 2024-07-12

**Soundness:** 3
**Presentation:** 3
**Contribution:** 3
**Rating:** 6
**Confidence:** 4

**Summary:**

This paper presents a multivariate forecasting model underlined by a dynamic spatial structure generation function, enabled by SVD-based parameterization and theoretically-bounded output space. Experiments on six commonly benchmark datasets in comparison to several existing baselines demonstrated the overall improved forecasting performance of the presented method as measured by MAE/RMSE/MAPE. Additional ablation studies further demonstrated the benefits of dynamic coefficient generation and the structured parameterization.

**Strengths:**

The concept of dynamic generation of spatial structure function is of novelty and has practical value for adapting to the varying spatial structure underlying time series data.

The SVD-based parameterization and low-rank approximation provides theoretical-based solutions to the challenge of identifying time-varying spatial structure functions.

The experiments considered a large number of common benchmarks and representative existing forecasting models. The performance was overall favorable, and the ablation study thorough.

The interpretability results in section 4.4 and Fig 2 are interesting, especially in uncovering the dynamic patterns underlying a dataset.

**Weaknesses:**

The obtained performance gain (Table 1) was overall marginal (up to 2-3 decimal points). The practical implication of such margin of improvements is not clear and need to be clarified. The effect of hyperparameters and random initialization on such margin of improvements also needs to be examined — adding statistics to the results over different random seeds of experiments is important.

In switching dynamic systems, it is also common to model the transition matrix over time as a linear combination of several global matrices, with time-varying mixing coefficients (without considering the proposed SVD decomposition and parameterization). It’d strengthen the paper to add discussion about the relation with this line of works, as well as experimental comparison.


Based on the results in Fig 3, it appears that increasing M from 1 to 2 in general introduced small difference in performance except in the Electricity dataset. This again raises some question on the significance of the dynamic spatial structure introduced in this paper. Stronger clarification on performance improvement and adding error bars to the results in Fig 3 will be important for verifying the contribution of the paper.

**Questions:**

Clarification on the observed margin of performance improvements and statistics/error bars to the results will be appreciated.

Better clarification and empirical results on relation with switching dynamics systems will be appreciated.

What is the computational cost difference in the ablation with or without common U-V basis (i.e., vs. w/ Um,Vm)?

**Limitations:**

The authors did not include sufficient discussion about the limitation or potential negative societal impact of the presented work.

---

> ### Author Rebuttal · Authors · 2024-08-04
>
> >Q1 The obtained performance gain (Table 1) was overall marginal (up to 2-3 decimal points). Adding statistics to the results over different random seeds of experiments is important.
> > Clarification on the observed margin of performance improvements and statistics/error bars to the results will be appreciated.
>
>
> Thank you for your comments and suggestions. We would like to point out that the performance gains of 2-3 decimal points are often considered to be **notable improvements in time series forecasting literature**, e.g., the prior works iTransformer [1] and Card [2] also achieve similar gains. To further demonstrate it more clearly, we provide the **improvements (percentage)** over the best baseline methods in the following table.
> | Dataset    | Electricity | Weather | PEMS| ETTh2 | Traffic | Solar |
> |:------:|:------:|:------:|:------:|:------:|:------:|:------:|
> | **Improvement**| 2.06%   | 3.97%   | 3.17%  | 1.06%   | 1.66%   | 2.15%  |
>
>  In summary, our proposed method achieves an **average performance improvement of 2.35%** over the best baselines. This enhancement is significant in the field of time series forecasting and verifies the effectiveness of our proposed model.
> Regarding the effect of random initialization, as mentioned in line 263 of our manuscript, our reported results are actually averaged over three runs with different random seeds. We will further add the statistics (standard deviation) into the tables as well as error bars in our revised manuscript. In addition, **the error bars are also provided in Figure 1 in the supplied PDF file**.
>
> [1] iTransformer: Inverted Transformers are Effective for Time Series Forecasting, ICLR 2024.
>
> [2] Card: Channel Aligned Robust Blend Transformer for Time Series Forecasting, ICLR 2024.
>
>
> >Q2 In switching dynamic systems, it is also common to model the transition matrix over time as a linear combination of several global matrices, with time-varying mixing coefficients. Better clarification and empirical results on relation with switching dynamics systems will be appreciated.
>
> Thank you for your suggestion. In switching dynamic systems, **the transition matrix is used to model the temporal dependencies**. The recent representative works include Hippo [1], LSSL [2], S4 [3], and Mamba [4]. In contrast, **the structured matrix basis is used to capture the spatial structures** in our proposed method. We have conducted the additional experiments to **compare with S4 in Table 2 in the supplied PDF file** and will also add the experimental results as well as the corresponding discussion in our revised manuscript.
>
> [1] Hippo: Recurrent Memory with Optimal Polynomial Projections, NeurIPS 2020.
>
> [2] Combining Recurrent, Convolutional, and Continuous-time Models with Linear State Space Layers, NeurIPS 2021.
>
> [3] Efficiently Modeling Long Sequences with Structured State Spaces, ICLR 2022.
>
> [4] Mamba: Linear-time Sequence Modeling with Selective State Spaces, arXiv 2023.
>
>
> >Q3 Based on the results in Fig 3, it appears that increasing M from 1 to 2 in general introduced small difference in performance except in the Electricity dataset.
> > Stronger clarification on performance improvement and adding error bars to the results in Fig 3 will be important for verifying the contribution of the paper.
>
> Thank you for your suggestion. To further demonstrate the effectiveness of our proposed dynamic spatial structure, we present the **improvements (percentage)** in the table below as $M$ increases from 1 to 2. The results show that our method achieves an **average performance improvement of 5.66%** across six different datasets. This significant improvement underscores the effectiveness of our dynamic spatial structure design. We also include the **error bars in Figure 2 in the supplied PDF file** to show these more clearly.
> | Dataset    | Electricity | Weather | PEMS| ETTh2 | Traffic | Solar |
> |:------:|:------:|:------:|:------:|:------:|:------:|:------:|
> | **Improvement**| 10.77%   | 5.45% | 3.10%| 2.20%  | 2.80%  | 9.63%|
>
>
>
>
> >Q4 What is the computational cost difference in the ablation with or without common U-V basis (i.e., vs. w/ Um,Vm)?
>
> Let $M$ be the dimension of the matrix basis, $N$ the number of nodes, $K$ the rank, and $D$ the feature dimension. As shown in Equation 10 of our manuscript, if each matrix basis has specific coordinates $\mathbf{U}_m$ and $\mathbf{V}_m$, then $\mathbf{V}_m^T \mathbf{x}$ and $\mathbf{U}_m \mathbf{y}^\prime$ will be calculated $M$ times. Consequently, the computational costs are $\mathcal{O}(MNKD)$ and $\mathcal{O}(NKD)$ for the models without and with a common $\mathbf{U}$-$\mathbf{V}$ basis, respectively. The computational cost difference will be included in our revised manuscript.
>
>
>
> >Q5 The authors did not include sufficient discussion about the limitation or potential negative societal impact of the presented work.
>
> Thanks for your suggestion. We will add more discussion on the limitation including, the model performance on **long-term forecasting tasks and hyperparameter selection**.

---

> > ### Comment · Reviewer_aCCB · 2024-08-08
> >
> > Thanks for the rebuttal effort -- I appreciate the new baseline and the error bars added to the pdf. It'd be highly recommended for the authors to add the error bar (or their numerical values) to the main text of the paper if accepted, as that's the best evidence for the significance of the margins of improvements (instead of stating that the range is common in literature). I will raise my rating.

---

> > > ### Author Response · Authors · 2024-08-09
> > >
> > > We sincerely thank the reviewer for raising the rating! We will include error bars in our revised manuscript.

---

### Official Review · Reviewer_ftFS · 2024-07-13

**Soundness:** 3
**Presentation:** 3
**Contribution:** 3
**Rating:** 6
**Confidence:** 3

**Summary:**

This paper presents a method to learn dynamic spatial structures in spatio-temporal forecasting tasks. Specifically, it proposes to parametrize the dynamic structures with a convex combination of fixed matrix bases, and the bases are further confined to be in the same coordinate system. Beyond that, it also imposes low-rank assumption on the coordinates to further reduce complexity. Empirical evidences show the proposed method achieve impressive accuracy in a number of spatio-temporal forecasting benchmarks, and the found spatial structures are highly interpretable.

**Strengths:**

- The paper is well-written with clear introduction of motivation behind major proposals and solid theoretical ground.
- Extensive experiments are provided to showcase the advantages of the proposed method empirically that addresses both efficacy and efficiency concerns.

**Weaknesses:**

The forecasting horizon in the experiments is quite different from the setting in some baselines, such as PatchTST and FEDformer. While long-term forecasting is not major claim of the paper, I wonder if the method scales well with prediction length.

**Questions:**

1. Why is TCN selected for temporal encoding?
2. Is there any issue that the learned bases degenerate to be trivial or converge to be close?

**Limitations:**

Limitations are discussed, and I agree that the selection of $M$, i.e. the number of bases, is highly empirical and ad-hoc.

---

> ### Author Rebuttal · Authors · 2024-08-04
>
> >Q1 The forecasting horizon in the experiments is quite different from the setting in some baselines, such as PatchTST and FEDformer. While long-term forecasting is not major claim of the paper, I wonder if the method scales well with prediction length.
>
>
> Thanks for your comments. Indeed, long-term forecasting is not the main focus of our proposed method, and the present version of Sumba is not aimed at addressing long-term predictions. But **we do conduct experiments with a forecasting horizon of 96 and as Table 1 in the supplied PDF file shows,  Sumba still gives rise to favorable performance** in comparison to PatchTST, FEDformer, and iTransformer which are designed specifically for long-term forecasting. We will leave it for our future work to further enhance its long-term forecasting capability and add this discussion to the limitation section.
>
> >Q2 Why is TCN selected for temporal encoding?
>
> Since the main focus of the paper is to effectively capture the spatial structures under the GCN framework, TCN is chosen for temporal encoding **due to its simplicity and can be easily integrated with the GCN operation**. In addition, **TCN architecture** (with appropriate modifications) [1] **also proved very effective in time series analysis** recently. We are also exploring the possibility of integrating our proposed spatial structure modeling with other temporal encoders such as Transformers, Structured State Space models, etc.
>
> [1] ModernTCN: A Modern Pure Convolution Structure for General Time Series Analysis, ICLR 2024.
>
> > Q3 Is there any issue that the learned bases degenerate to be trivial or converge to be close?
>
> **We do not empirically run into this issue** in our experiments (conducted on six public datasets from various domains). We hypothesize that **the degenerated issue may arise when the spatial structures are static** and can be represented by a single graph structure. In such a case, the model would only attend to one of the matrices in the basis (that is, all weights are concentrated in one entry while the others are zeros in the coefficient $\alpha$). This will not have a negative impact on the model's performance.
>
> > Q4 Limitations are discussed, and I agree that the selection of  $M$, i.e. the number of bases, is highly empirical and ad-hoc.
>
> Yes, it will be our future work to explore the adaptive selection of $M$ as well as enhancing the model’s long-term forecasting capability.

---

### Author Rebuttal · Authors · 2024-08-04

We sincerely thank all reviewers for their precious comments and valuable suggestions. Our major points of view are summarized as follows.

- We conducted **additional experiments with a forecasting horizon of 96**.
- We provide explanations on **the choice of temporal encoding function and the potential degeneration issue**.
- We clarify the **performance improvements** and include the error bars to show them more clearly.
- We include **S4 as a new baseline method** and the corresponding discussion is also presented.
- We provide the **computational cost** of the models with/without a common $\mathbf{U}$-$\mathbf{V}$ basis.
- We further clarify the **implementation details** and add a more **detailed analysis and comparison with graph-based spatiotemporal models**.

---

### Decision · Program_Chairs · 2024-09-25

**Decision:**

Accept (poster)

**Comment:**

Thanks for your submission to NeurIPS! The reviewers have reach a
consensus, and it is clear that the reviewers are positive with its
quality. The recommended decision from the AC is accept. The authors may
take into account the feedbacks of the reviewers to further improve the
paper. There could be open or extension questions that may not be easy
to answer while possible comments (probably in the supplemental) could
help.

Bests

The AC